# Spatio-temporal mRNA tracking in the early zebrafish embryo

Karoline Holler [1], Anika Neuschulz [1], Philipp Drewe-Boß[1], Janita Mintcheva[1], Bastiaan Spanjaard[1], Roberto Arsiè [1], Uwe Ohler [1,2], Markus Landthaler [1,3] & Jan Philipp Junker [1✉]

Early stages of embryogenesis depend on subcellular localization and transport of maternal mRNA. However, systematic analysis of these processes is hindered by a lack of spatio-temporal information in single-cell RNA sequencing. Here, we combine spatially-resolved transcriptomics and single-cell RNA labeling to perform a spatio-temporal analysis of the transcriptome during early zebrafish development. We measure spatial localization of mRNA molecules within the one-cell stage embryo, which allows us to identify a class of mRNAs that are specifically localized at an extraembryonic position, the vegetal pole. Furthermore, we establish a method for high-throughput single-cell RNA labeling in early zebrafish embryos, which enables us to follow the fate of individual maternal transcripts until gastrulation. This approach reveals that many localized transcripts are specifically transported to the primordial germ cells. Finally, we acquire spatial transcriptomes of two *xenopus* species and compare evolutionary conservation of localized genes as well as enriched sequence motifs.

[1] Berlin Institute for Medical Systems Biology, Max Delbrück Center for Molecular Medicine, Berlin, Germany. [2] Department of Biology, Humboldt University, Berlin, Germany. [3] IRI Life Science, Institute of Biology, Humboldt University, Berlin, Germany. ✉email: janphilipp.junker@mdc-berlin.de

During embryonic development, initially pluripotent cells differentiate into a multitude of different cell types with distinct gene expression programs and spatial organization. In many species, the earliest stages of development depend strongly on RNA transport and intracellular localization of maternal transcripts[1,2]. Investigating these processes with transcriptome-wide methods remains challenging. Advances in single-cell RNA sequencing (scRNA-seq)[3–6] have made it possible to reconstruct developmental differentiation trajectories by pseudo-temporal ordering of single-cell transcriptomes[7–10]. However, these approaches do not yield the information about RNA localization and transport. Therefore, methods that combine spatially resolved transcriptomics and tracking of maternal transcripts are needed to investigate RNA transport in early development.

While many methods for spatially resolved transcriptomics have emerged in recent years[11,12], state-of-the-art spatial RNA-seq methods typically have not reached the necessary resolution (single-cell level) yet[13]. This limitation also holds for the tomo-seq method[14,15], in which RNA from serial cryosections is extracted, barcoded, and sequenced. Microscopy-based approaches using sequential fluorescent in situ hybridization hold great promise for spatial transcriptomics with sub-single-cell resolution[16–18], but application of these methods to early embryos is technically challenging. Similarly, methods based on proximity labeling, which are powerful approaches for determining the transcriptome associated with different cellular compartments, require specific markers and transgenic engineering, and have not been successfully applied to early vertebrate embryos yet[19,20].

The temporal aspect of RNA expression is, by nature of the experiment, even harder to catch in single cells. While live microscopy based on fluorescent reporters is well established, methods for live measurement of transcript abundance typically consider only a couple of genes and are difficult to apply in live multicellular animals[21]. However, a cell's "future transcriptome" can, within certain limits, be inferred from RNA-seq data by counting the occurrence of intronic reads[22]. Recent methods allow direct measurement of the transcriptional history of single cells in cell culture by introducing modified nucleotides into newly synthesized RNA[23–28]. However, this approach has not been established in live embryos yet.

In this work, we use a combination of spatially resolved transcriptomics and RNA labeling to study the spatio-temporal transcriptome during the first few hours of the zebrafish development. Specifically, we improve the tomo-seq method to measure RNA localization in one-cell stage zebrafish embryos with high spatial resolution. We use this information to systematically identify genes with subcellular localization patterns within the one-cell stage embryo. Furthermore, we develop a protocol for single-cell RNA labeling in early zebrafish embryos. This approach enables us to follow the fate of individual maternal transcripts until gastrulation, and thereby deduce to which cell types the localized transcripts contribute in embryonic development. We additionally investigate mRNA localization in an evolutionarily related system, oocytes from *Xenopus laevis* and *Xenopus tropicalis*. In summary, this data allows us to derive principles of mRNA localization in vertebrate oocytes, as well as evolutionary conservation and enriched sequence motifs.

## Results

**Tomo-seq in one-cell stage embryos**. For a systematic investigation of spatial RNA gradients in the zebrafish one-cell stage embryo, we established an enhanced, more sensitive version of the tomo-seq method[14] ("Methods" section): we embedded and

oriented individual embryos at the one-cell stage (~30 min after fertilization) along the microscopically visible animal–vegetal axis. We then sectioned the cell and the yolk sac into 96 sections (Fig. 1a) and followed the tomo-seq protocol ("Methods" section) for a total of three independent samples. We found that the majority of the mRNA is located in the blastodisc, which is positioned adherent to the yolk sac at the animal pole of the embryo (Fig. 1b and Supplementary Fig. 1). To account for this pattern, we normalized transcript counts by total UMI counts per section, and recovered known localization patterns, as shown for important patterning genes like *dazl*, *trim36*, *grip2a*, *wnt8a*, and *celf1* (Fig. 1c). We found that our tomo-seq library has high complexity, which enabled us to confidently determine spatial expression patterns of a large number of genes: we found an average of 13.4 M unique transcripts (UMIs) per sample, and we observed that at the chosen sequencing depth (61 M reads on average), we are still far from reaching saturation, as determined by comparing UMI counts to read counts (Fig. 1d). Gene expression of individual replicates correlates well ($R = 0.99$, Fig. 1e and Supplementary Fig. 1).

**Systematic identification of mRNA localization patterns**. In order to identify gene expression patterns in a systematic way, we clustered our spatial gene expression data based on a self-organizing map[29], which sorted the cumulative gene expression traces along a linear axis of 50 profiles (Supplementary Dataset 1). As a result, we found three major groups of localized mRNA (Fig. 2a and Supplementary Fig. 2): one localized to the animal side in profiles 1–8, one group of genes that was more or less equally distributed across all sections, and a third group of genes that was localized to the most vegetal part of the yolk sac in profiles 48–50. While the first group is likely an overlap between genes that had been localized to the animal pole before fertilization and transcripts that are transported by non-specific cell-directed cytoplasmic streaming upon fertilization[30], the third group corresponds to a distinct set of transcripts that are specifically transported and retained at the vegetal pole[31].

Since vegetally localized genes have been reported to play major roles in early development, especially in germ cell development and dorsoventral axis specification[32,33], and since this group of genes exhibited the most pronounced and reproducible spatial pattern in the one-cell stage embryo (Supplementary Fig. 2), we decided to investigate it in more depth. We compared vegetally localized genes in profiles 48–50 between three replicates and found an overlap of 66 genes (Fig. 2b). A subset of the localized genes was not detected in one of the replicates (Fig. 2c), which was likely due to the overall low expression of these genes. Another subset of genes that was defined as vegetally localized in one sample, was detected just below the threshold, in profiles 46 and 47, in another replicate (Supplementary Fig. 2). Since manual inspection revealed that these genes had expression traces similar to genes previously annotated as vegetal (examples in Supplementary Fig. 2), we decided to demand a vegetally localized gene to be in profiles 46–50 in all replicates, and in profile 48–50 in at least one of the replicates. With these criteria, we defined 97 genes to be localized vegetally, which increases the number of known vegetal genes by about tenfold (see Supplementary Dataset 2 for genes and references). Moreover, this list includes all genes that to our knowledge have previously been shown to localize vegetally. We validated seven genes from this list, together with the animally localized gene *exd2*, by whole-mount in situ hybridization (Fig. 2d). In summary, tomo-seq allowed us to determine subcellular RNA localization in the one-cell stage zebrafish embryo on the transcriptome-wide level, which led to the

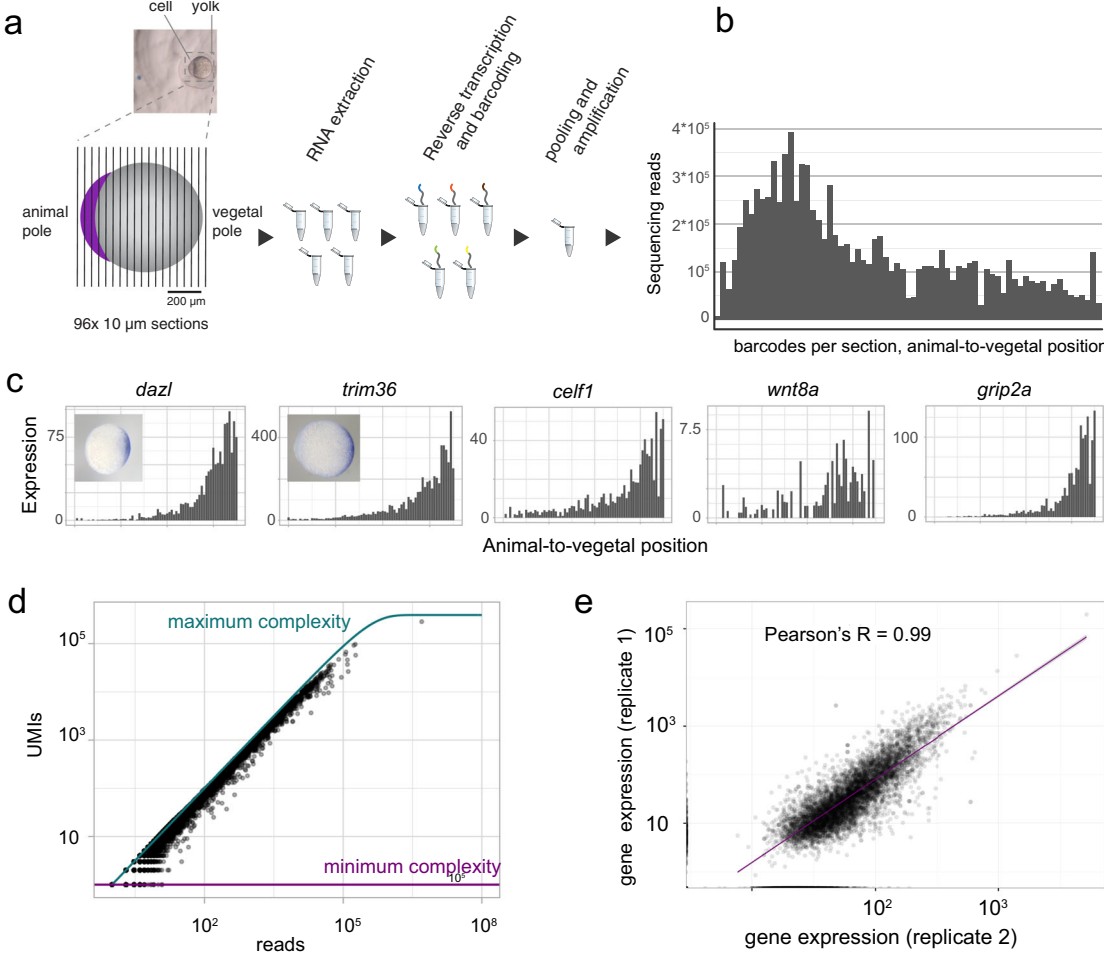

**Fig. 1 Tomo-seq in one-cell stage embryos. a** Experimental outline: the embedded embryo is cryosectioned into 96 slices that are put into separate tubes. After adding spike-in control RNA, RNA is extracted. In a reverse transcription step, spatial barcodes are introduced. Samples are then pooled and amplified by in vitro transcription and a final library PCR. Scale bars are 200 μm. **b** Histogram shows raw transcript counts per section. **c** Tomo-seq tracks for the known vegetally localized genes *dazl*, *trim36*, *celf1*, *wnt8a*, and *grip2a* and whole-mount in situ hybridizations for *dazl* and *trim36*. **d** Sequencing depth, shown as UMI saturation per gene. Maximum complexity is determined as by Grün et al.[73]. **e** Correlation of two tomo-seq experiments (total counts summed over all sections). Line is a linear fit to the data. The difference in scale between the two axes is caused by differences in sequencing depth between the two replicates.

identification of 97 genes that are specifically localized at the vegetal pole.

**Single-cell RNA labeling in early zebrafish embryos.** To better understand the role of the vegetally localized genes in early development, it is important to follow the fate of maternal transcripts over time, in order to find out to which cell types they later contribute. Since the vegetal pole is an extraembryonic position, it is not clear a priori where the vegetal transcripts are later transported. The first major embryonic cell type decisions occur at gastrulation, which in the zebrafish happens at ~6 h post fertilization (h.p.f.)[34]. Zygotic transcription starts at ~3 h.p.f., and gastrulation stages are characterized by a coexistence of maternal and zygotic transcripts. It is therefore crucial to distinguish maternal transcripts of localized genes from zygotic expression of the same genes. We hence decided to develop an approach to distinguish maternal and zygotic transcripts transcriptome-wide and on the single-cell level. Our method is based on single-cell RNA metabolic labeling (scSLAM-seq[23]), which enables us to distinguish maternal and zygotic transcripts by incorporation of the nucleotide analog 4-thiouridine (4sU). After a chemical derivatization step using iodoacetamide (IAA), labeled uridines

are detected as T-to-C mutations upon sequencing[35] (Fig. 3a). Several approaches for RNA labeling in single cells have been introduced recently[23–28]. However, these approaches are limited to cultured cells and have not been applied to live vertebrate embryos yet. Furthermore, they are mostly plate-based and (with the exception of Qiu et al.[27]) not compatible with high-throughput scRNA-seq by droplet microfluidics. In order to study embryonic development, and to also capture rare cell types, such as germ cells, it was crucial to overcome these limitations. We therefore developed a scSLAM-seq protocol that does not require cell lysis prior to IAA derivatization, which allowed us to load intact cells for droplet microfluidics scRNA-seq (Fig. 3a and "Methods" section). To do so, cell membranes are permeabilized for IAA uptake by methanol fixation (Supplementary Fig. 3). Compared to cultured cells, a major challenge in live embryos is to deliver the labeling reagent into the cells. Indeed, we found that the addition of 4sU into the water did not yield high labeling efficiencies (Supplementary Fig. 3). In bulk experiments, injection of 4-thiouridine-triphosphate (4sUTP) into one-cell stage zebra-fish embryos has been used successfully for studying maternal-to-zygotic transition[36]. Using the triphosphate 4sUTP has the additional advantage that the nucleotide analog is available immediately for incorporation into RNA without relying on

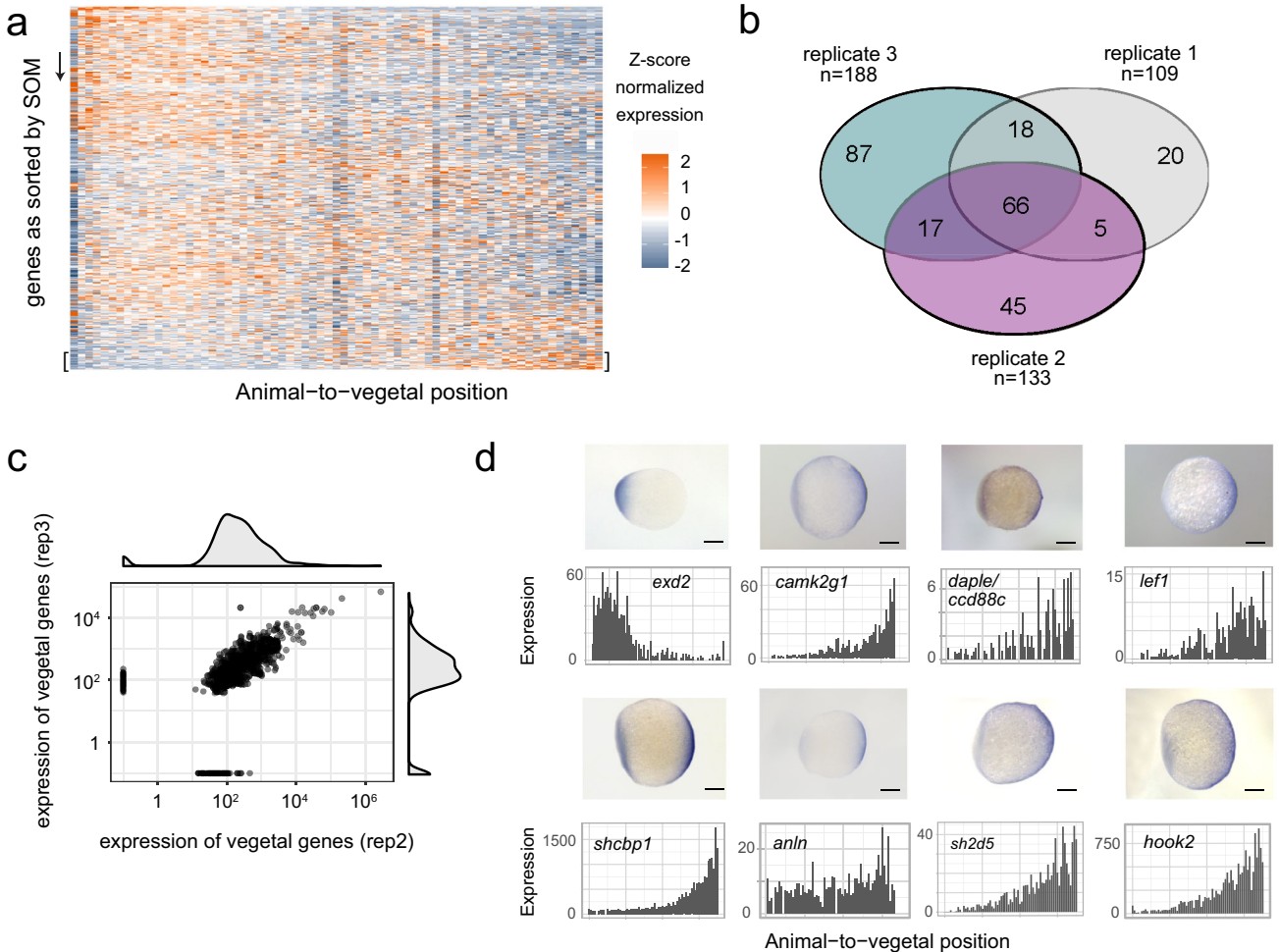

**Fig. 2 Systematic identification of mRNA localization patterns. a** Heatmap representation of *z*-score normalized expression per section in a zebrafish one-cell stage embryo. Genes on the *y*-axis as sorted into profiles 1–50 by SOM (self-organizing map), spatial position in the embryo on the *x*-axis. **b** Vegetally localized genes per sample (profiles 48–50). **c** Expression correlation of vegetally localized genes between two replicates. Genes on the axes are only detected in one sample. **d** Comparison of tomo-seq and whole-mount in situ hybridization for selected newly described vegetally localized genes, as well as the animally localized gene *exd2*. Scale bars are 200 μm.

further metabolic conversion. We observed efficient RNA labeling and successful conversion with IAA on bulk RNA upon 4sUTP injection (Supplementary Fig. 3). We then proceeded to prepare single-cell suspensions at 50% gastrulation, fixed the cells with methanol, and converted 4sUTP to a cytosine analog in intact cells (Fig. 3a and Supplementary Fig. 3). We then sequenced a total of 7472 cells with 10× Genomics Chromium, and analyzed the data with custom code (see "Methods" section). Comparison of mutation rates in 4sUTP injected embryos with control samples confirmed that the T-to-C mutation rate is increased strongly and specifically (Fig. 3b). We found that the 4sUTP treatment resulted in a bimodal distribution of the T-to-C mutation frequency per gene (Fig. 3c), suggesting a good separation of labeled and unlabeled reads. The observed labeling efficiency of 5% corresponds to a low false negative rate of ~1% of unlabeled zygotic transcripts ("Methods" section), which demonstrates that we can reliably distinguish maternal and zygotic transcripts.

Unsupervised clustering of cells, using the information of the labeled mRNA, resulted in eight cell clusters (Fig. 3d) with defined marker gene expression (Fig. 3e, and Supplementary Datasets 3 and 4). We then clustered cells based on their unlabeled mRNA (Fig. 3d), and imposed cell identities as defined based on labeled mRNA. As expected, clustering based on unlabeled (maternal) mRNA separated cell types much less than

clustering on labeled (zygotic) RNA, with the notable exception of the enveloping layer and the primordial germ cells (PGCs). These two cell types had the most distinct marker gene signature (Fig. 3e and Supplementary Dataset 3), and the cells of the enveloping layer were characterized by a particularly high labeling rate (Fig. 3f), which indicates high transcriptional or proliferative activity. The PGCs, on the other hand, display the lowest labeling rates among all cells at this developmental stage (Fig. 3f), in agreement with reports that show very slow increase of prospective PGCs before gastrulation[37,38].

**Tracking the fate of maternal transcripts by scSLAM-seq.** Next, we set out to assess if any of the maternal, vegetally localized genes were overrepresented in specific cell types. At 6 h.p.f., we still detected unlabeled RNA for 91 of the 97 genes that were localized at the one-cell stage. We filtered out lowly expressed genes, and for the remaining 47 genes, we calculated the expression fold change for each of the cell types compared to all other cell types (Fig. 4a and Supplementary Fig. 4). We found that the vegetally localized genes were significantly enriched in PGCs ($p$ value $= 4.67 \times 10^{-5}$), with 28 of them being marker genes of that particular cluster (Supplementary Dataset 4). The logarithmic fold enrichment of vegetal genes in PGCs follows a bimodal

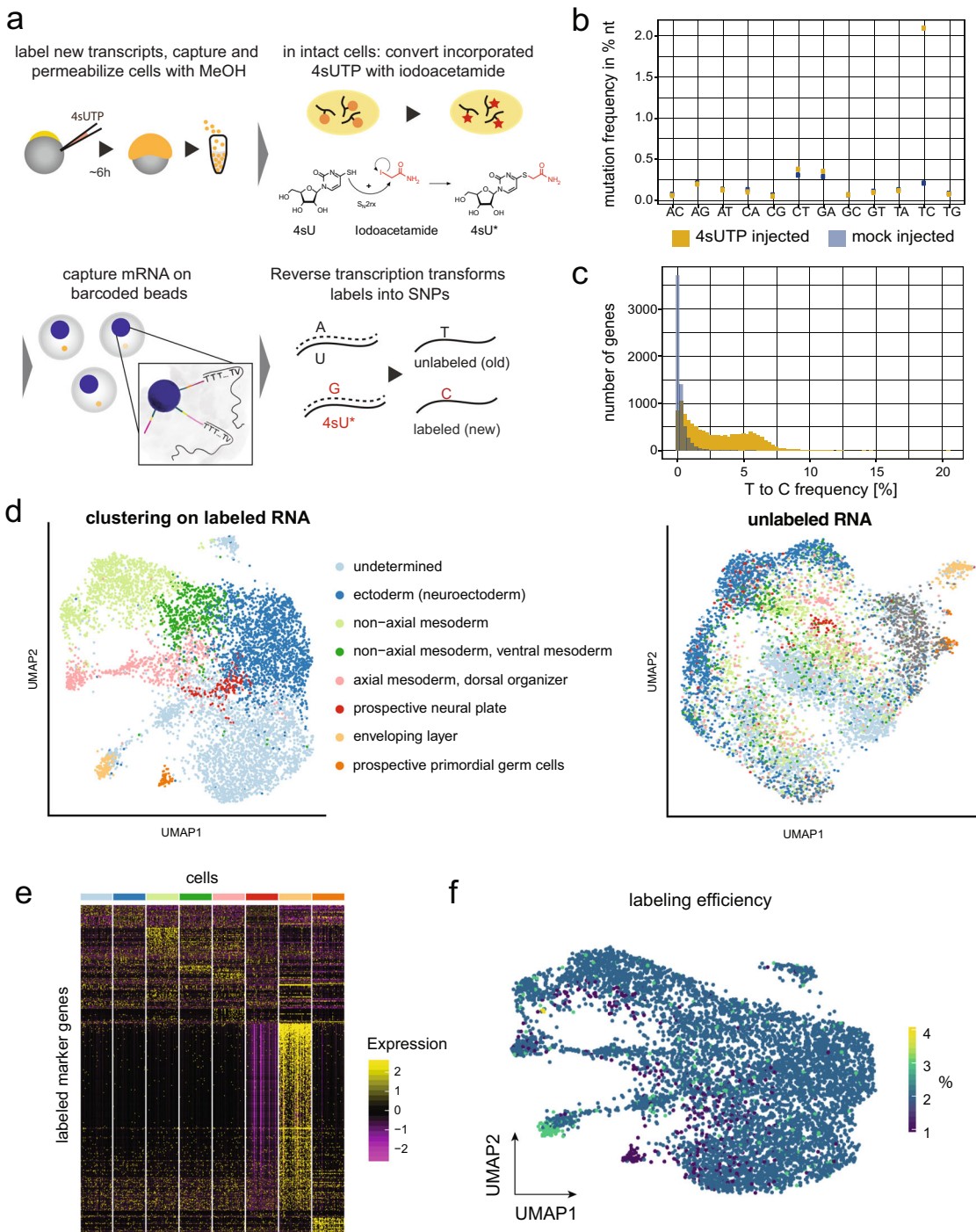

**Fig. 3 Single-cell RNA labeling in early zebrafish embryos. a** Schematic representation of the protocol: 4sUTP (4-thiouridine-triphosphate) injection into zebrafish one-cell stage embryos, dechorionation, dissociation into single cells at gastrulation stage, and MeOH (methanol) fixation (see "Methods" section). Incorporated 4sUTP is converted in a SN2 reaction with iodoacetamide into a cytosine analog. The single-cell solution is then loaded onto 10× Genomics Chromium, and chemical labels lead to T-to-C conversions during reverse transcription. **b** Nucleotide mutation frequencies of a scSLAM-seq library after injecting 4sUTP or Tris and quality filtering of the data. **c** Histogram of T-to-C mutations in 4sUTP- and Tris-injected embryos. **d** UMAP representation of cells based on labeled RNA (left side) and unlabeled RNA (right side). For the latter, we imposed cell identities as determined on the basis of labeled RNA. **e** Marker gene expression of labeled cells in different cell types (color code as in **d**). Cell number per cluster was downsampled to equal numbers. **f** Transcript labeling efficiency in single cells in percent, projected on the UMAP representation for labeled RNA.

distribution (Fig. 4b, dashed line), suggesting two subpopulations of vegetal genes. Indeed, we can deconvolve the bimodal distribution into two normal distributions, where one resembles the distribution of randomly sampled genes (Fig. 4b light blue and gray), while the other has a significantly higher mean fold change

(Fig. 4b dark blue, $p$ value $= 1.7 \times 10^{-4}$), suggesting a role of these genes in germ cell specification or development. We show the average expression at 6 h.p.f. for some of the new candidates (*sh2d5*, *itpkca*, *ndel1b*, *anln*, *krtcap2*, and *ppp1r3b*) in Fig. 4c, next to the remaining maternal expression of well-established germ

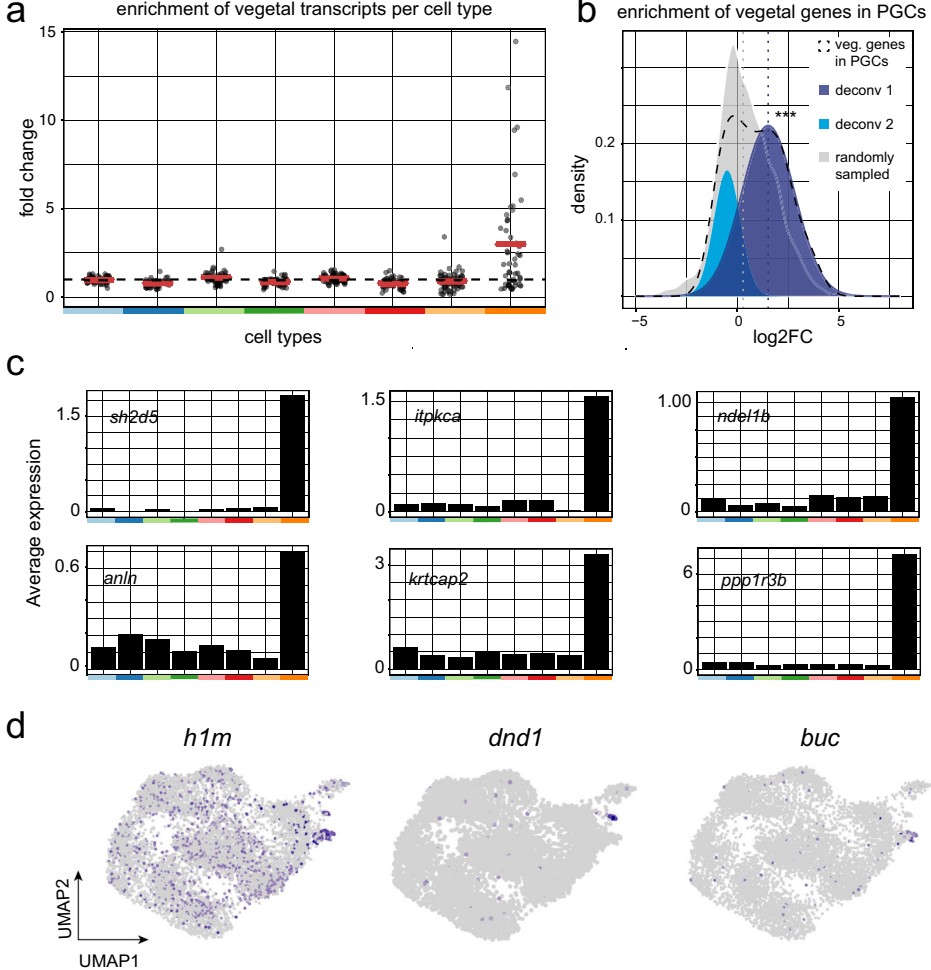

**Fig. 4 Tracking the fate of maternal transcripts by scSLAM-seq. a** Fold change enrichment of maternal vegetally localized genes in the different cell types versus all other cells (color code as in Fig. 3d). Genes with an average expression <0.1 transcripts/cell were excluded from this analysis. Red bars represent mean values. **b** Deconvolution of the bimodal distribution of vegetally localized genes in PGCs (black dashed line) into two normal distributions (light and dark blue). The mean value of the dark blue distribution is significantly higher than that of a randomly sampled distribution ($m_{gray} = 0.4$, $m_{dark\ blue} = 1.52$, $p$ value $= 1.7 \times 10^{-4}$, Welch's $t$ test, one-sided). **c** Average expression of most highly enriched genes in PGCs in different cell types (color code as in Fig. 3d). **d** Unlabeled RNA expression of established germ cell markers on a UMAP representation.

cell factors (Fig. 4d). In summary, our scSLAM-seq analysis revealed that a large number of the vegetally localized transcripts are later transported to PGCs, thereby allowing us to identify a set of novel candidate genes with a potential function in germ cell specification and differentiation.

While the main goal of our scSLAM-seq analysis was to track the vegetally localized maternal genes, our dataset can also give insight into all other maternal genes. We therefore expanded the enrichment analysis in Fig. 4a from vegetally localized genes to all maternal transcripts. We observed that the majority of maternal transcripts are not enriched in any specific cell type, but for a small fraction of genes we found an enrichment of maternal transcripts in PGCs, and to a lesser degree also in enveloping layer cells (Supplementary Fig. 5).

Furthermore, we investigated the potential involvement of maternal factors in cell–cell interactions. We first identified ligand–receptor pairs between cell types in our scSLAM-seq data, using two different computational methods (CellChat and CellPhoneDB, see "Methods" section). In both approaches, we found that none of the remaining 47 vegetal genes are involved in annotated ligand–receptor interactions, suggesting that the role of maternal vegetal genes is mostly related to cell specification (in particular specification of PGCs), and not so much to cell–cell

communication. When analyzing all genes, CellChat identified several potential cell–cell interactions. However, expression of the involved ligands and receptors was mostly zygotic, and we did not detect any interactions between PGCs and other cell types (Supplementary Fig. 5). While the results of our CellPhoneDB analysis were generally similar, we additionally also observed several maternal ligands and receptors in PGCs, with potential interaction partners in a variety of cell types (Supplementary Fig. 6). This discrepancy between the two computational approaches is probably due to differences in the underlying ligand–receptor databases.

**Evolutionary conservation of vegetal mRNA localization**. We next decided to determine the conservation of germ cell factors by comparing vegetally localized genes in zebrafish and *Xenopus*. Our choice of *Xenopus* was motivated by reports showing that 3′ untranslated region (UTR) sequences of a zebrafish germ plasm gene can drive transcript localization in frog oocytes[39], and furthermore that the localization machineries of two different *Xenopus* species, *X. tropicalis* and *X. laevis*, are functionally overlapping[39,40], which suggests that RNA localization is driven by common *cis*-regulatory elements. *Xenopus* as well as

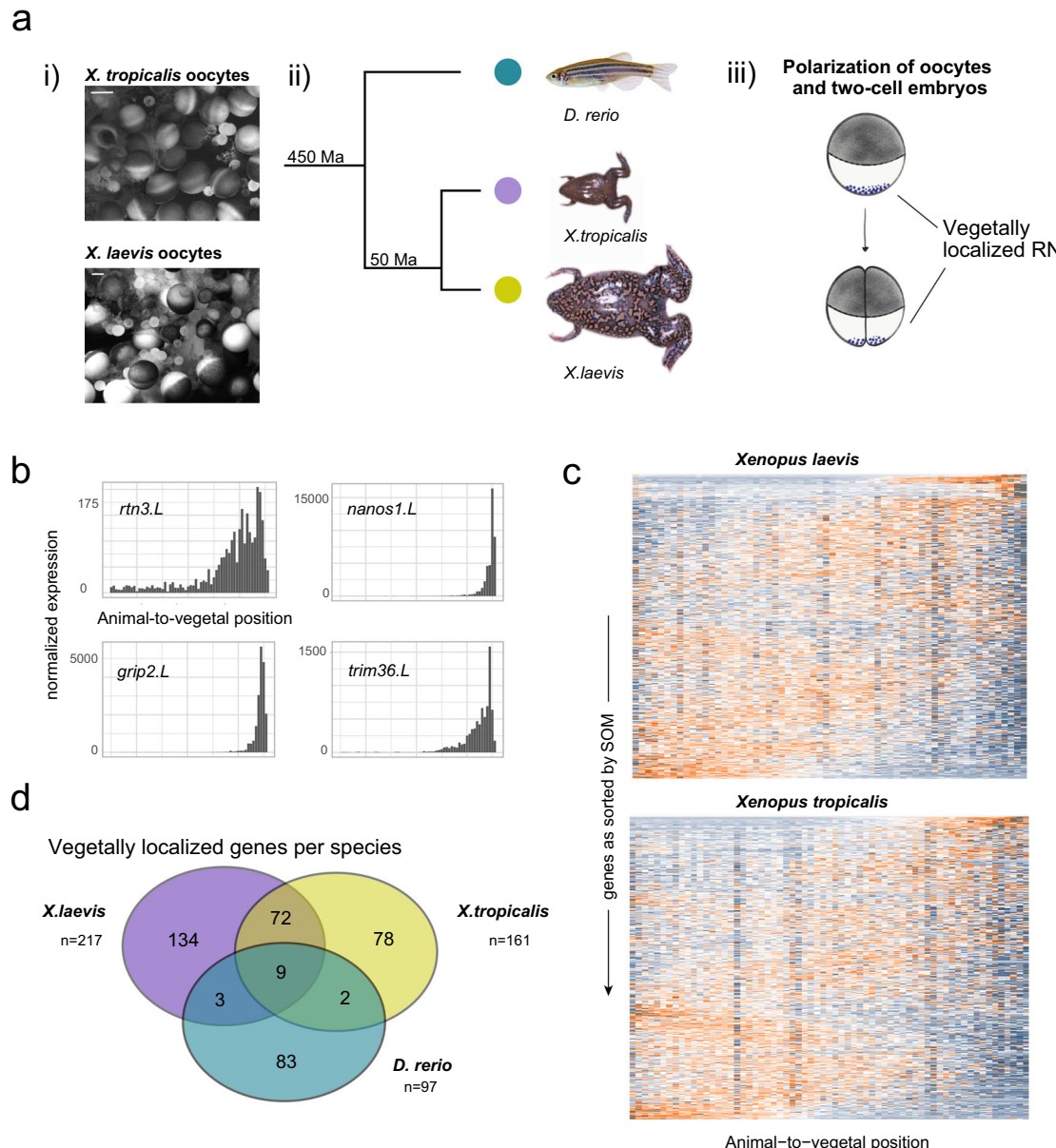

**Fig. 5 Evolutionary conservation of vegetal mRNA localization. a** (i) Light microscopy view of whole oocyte lobes from *X. laevis* and X. *tropicalis* before dissociation for one of the two replicates. Scale bar are 500 μm. (ii) Phylogenetic distance of *Xenopus* species and zebrafish as described in ref. [74] (Ma: million years). (iii) Deposition of germ plasm and dorsal factors (as purple dots) in *Xenopus* oocytes and after first cell division. **b** Tomo-seq tracks of vegetally localized genes *rtn3.L*, *nanos1.L*, *grip2.L*, and *trim36.L* in *X. laevis*. **c** Heatmap of z-score normalized expression per section in *Xenopus* oocytes. Genes on the y-axis as sorted into profiles 1–50 by SOM (self-organizing map), spatial position on the x-axis. **d** Overlap of vegetally localized genes in zebrafish and *Xenopus* species, considering only genes that were expressed in all three species at the respective developmental stage.

zebrafish use the vegetal pole to store factors for germ cell specification and dorsoventral axis determination[41,42], which additionally suggests functional similarity despite a considerable evolutionary distance (Fig. 5a). However, it is important to point out that there are major structural differences between zebrafish and *Xenopus* embryos: zebrafish have one big yolk cell, with all blastomeres positioned at the animal pole. In contrast to this, the yolk cell is subdivided during cell division in *Xenopus*. Since existing *Xenopus* datasets are derived from pooled samples and do not provide a comparable spatial resolution[43–45], we decided to produce tomo-seq datasets of mature oocytes from *X. tropicalis* and *X. laevis*, with two replicates for each species (10 μm resolution for *X. tropicalis*, and 16 and 18 μm resolution for *X. laevis*).

After excluding lowly expressed genes and normalizing to the same number of transcripts per section, we recovered known localization patterns for important developmental factors (Fig. 5b). As before, we calculated cumulative expression patterns and clustered them with self-organizing maps (SOM; Fig. 5c, Supplementary Fig. 7, and Supplementary Datasets 5 and 6). In *X. tropicalis*, we found 151 genes to be localized animally (1.5%) and 161 to be localized vegetally (1.6 %), for *X. laevis* we identified 245 genes localized to the animal pole (1.9%) and 216 genes to the vegetal pole (1.7%; genes in Supplementary Dataset 7). In accordance with a previous study[43], the interspecies overlap of localized genes was relatively low for these two closely related frog species—30% for animally localized genes and 50% for vegetally localized genes. One important difference between high-

resolution tomo-seq data and earlier studies of *X. laevis* and *X. tropicalis* is the identification of a distinct group of animally localized genes and their corresponding motifs (Supplementary Fig. 7). The existence of animally localized genes was previously controversial, since either very few (0.2%, Owens et al.[44]) or a large majority of genes (94.4%, Sindelka et al.[45]) were found to be enriched at the animal pole. This highlights the advantages of our robust high-resolution analysis of subcellular RNA localization. However, it is important to note that differences in data normalization and cutoff values also contribute to the different conclusions drawn in these studies.

While the overlap of the vegetal genes between the two *Xenopus* species with 81 genes was considerable, we only found nine genes to localize vegetally in all three species (Fig. 5d), showing a surprisingly variable transcript composition at the vegetal pole given the reported high degree of conservation of the localization machinery[39]. For comparison, 53 of the 97 vegetally localized zebrafish genes were detected in both *Xenopus* species above the expression cutoff. However, this analysis allowed us to propose that these nine genes, including known factors like *dazl* and *syntabulin*, but also less well-characterized genes like *camk2g1* and *ppp1r3b*, have a conserved function in germ cell development or dorsoventral axis development. In our scSLAM-seq data (Fig. 4), we found that three of these nine genes were still detected above our expression cutoff at gastrula stages. These three genes, *camk2g1*, *ppp1r3b*, and *dazl*, were all found to be enriched in the PGCs, indicating a conserved role in germ cell specification. Of note, *anln* is PGC enriched and localized in zebrafish, and it is vegetally localized in *X. tropicalis*, but unlocalized in *X. laevis*. The 3′UTR of *X. laevis anln* has a 1 kb long deletion, suggesting a functional contribution of that sequence to the localization (Supplementary Fig. 7). Table 1 gives an overview of the nine genes with conserved localization, their described cellular function (Xenbase.org, zfin.org) and the protein class of the translated product.

**3′UTR characteristics of vegetally localized genes**. Intracellular transcript localization is driven by *cis*-regulatory localization elements, present mainly in 3′UTRs of RNA molecules[39,46–49]. However, the exact nature of the localization motifs in early embryos has largely remained elusive. We therefore reasoned that our transcriptome-wide datasets of mRNA localization in three species might now open the door toward a more systematic analysis of these sequence elements. To this end, we decided to investigate shared sequence features of vegetally localized genes. Since tomo-seq detects only 3′ ends of transcripts, we performed bulk RNA-seq of one-cell stage zebrafish embryos in order to computationally identify expressed isoforms[50] ("Methods" section). In total, we detected 216 expressed isoforms of vegetally localized genes in zebrafish. We found that the 3′UTR sequences of vegetally localizing genes are on average 1.7-fold longer than for the background (*p* value $< 2.2 \times 10^{-16}$; Fig. 6a). In contrast to

this, we found only moderate differences in length of coding sequences (Fig. 6b) and expression level (Fig. 6c), and no differences in GC content of 3′UTRs (Supplementary Fig. 8). Longer 3′UTRs of vegetally localized genes could reflect complex cellular regulation of these transcripts with regard to localization and anchoring to the cytoskeleton, but could also be at least partially related to other regulatory processes, such as translational activity and RNA stability[51]. Finally, we searched for common *cis*-regulatory motifs by performing a *k*-mer enrichment analysis[52] of the 3′UTRs (Fig. 6d and "Methods"). We detected variations of a CAC core, several motifs containing a GUU sequence that has not been described yet, and a polyU stretch that was previously linked to increased RNA stability[53].

We next performed a *k*-mer enrichment analysis for the two *Xenopus* species by using the longest annotated 3′UTR isoform (Fig. 6e and Supplementary Fig. 8). In accordance with previous studies[46,54,55], and similar to our results for zebrafish, we found an enrichment of CAC-containing motifs in vegetally localized genes. We found the same polyU-motif as in zebrafish data, suggesting a conserved role in stability of maternal RNA. In *X. tropicalis*, we also found a motif consisting of the same GUU core we identified in zebrafish (Fig. 6d); however, the respective local sequence environment differed. In summary, we found a relatively high conservation of 3′UTR sequence motifs, which contrasts with the rather low conservation of vegetally localized genes that we observed in Fig. 5.

## Discussion

We here established improved versions of two methods, the tomo-seq approach for spatially resolved transcriptomics and single-cell SLAM-seq for RNA labeling. In tomo-seq, we achieved sub-single-cell resolution in zebrafish embryos at the one-cell stage. We observed that the complexity of the tomo-seq libraries was not a limiting factor, suggesting that our approach may be applicable to even smaller samples containing less mRNA. However, it is important to note that fertilized zebrafish eggs are very large cells (~700 μm), and sub-single-cell spatially resolved transcriptomics by tomo-seq would be much more challenging for smaller cells. The tomo-seq method is well suited for spatial transcriptomics in one-cell stage zebrafish embryos, since we expect the most striking patterns along the animal–vegetal axis. However, more complex spatial patterns, including, e.g., radial geometry, would be difficult to detect with our approach, which is based on serial sections in 1D. Hence, different strategies would be required to reveal the full spatial organization of the transcriptome in 3D. While sequencing-based approaches for spatially resolved transcriptomics in tissue sections typically do not reach the spatial resolution required here[13], methods based on sequential fluorescent in situ hybridization[16] have the potential to reveal more complex spatial patterns than can be detected by tomo-seq. However, analysis of a large number of serial sections

**Table 1 Vegetally localized genes in zebrafish, *X. laevis* and *X.tropicalis*.**

| Gene name (zebrafish) | Biological function | Protein function |
|---|---|---|
| *dazl* | Germ plasm component, translational activator | 3′UTR RNA binding |
| *sybu* | Dorsal/ventral axis specification | Kinesin binding |
| *grip2a* | Cytoskeleton organization, germ plasm | Receptor interaction |
| *rfn41* | E3 ubiquitin ligase | RING finger proteins |
| *rnf38* | Germ cell development in *X. laevis*[44] | RING finger proteins |
| *trim36* | Regulation of cell cycle | RING finger proteins |
| *ppp1r3b* | Glycogen metabolism | Phosphatase |
| *ctdsplb* | Regulation of RNA Pol II transcription | Phosphatase |
| *camk2g1* | Expressed in gut, nervous system, neural tube, involved in differentiation of inner ear[75] | $Ca^{2+}$-dependent kinase |

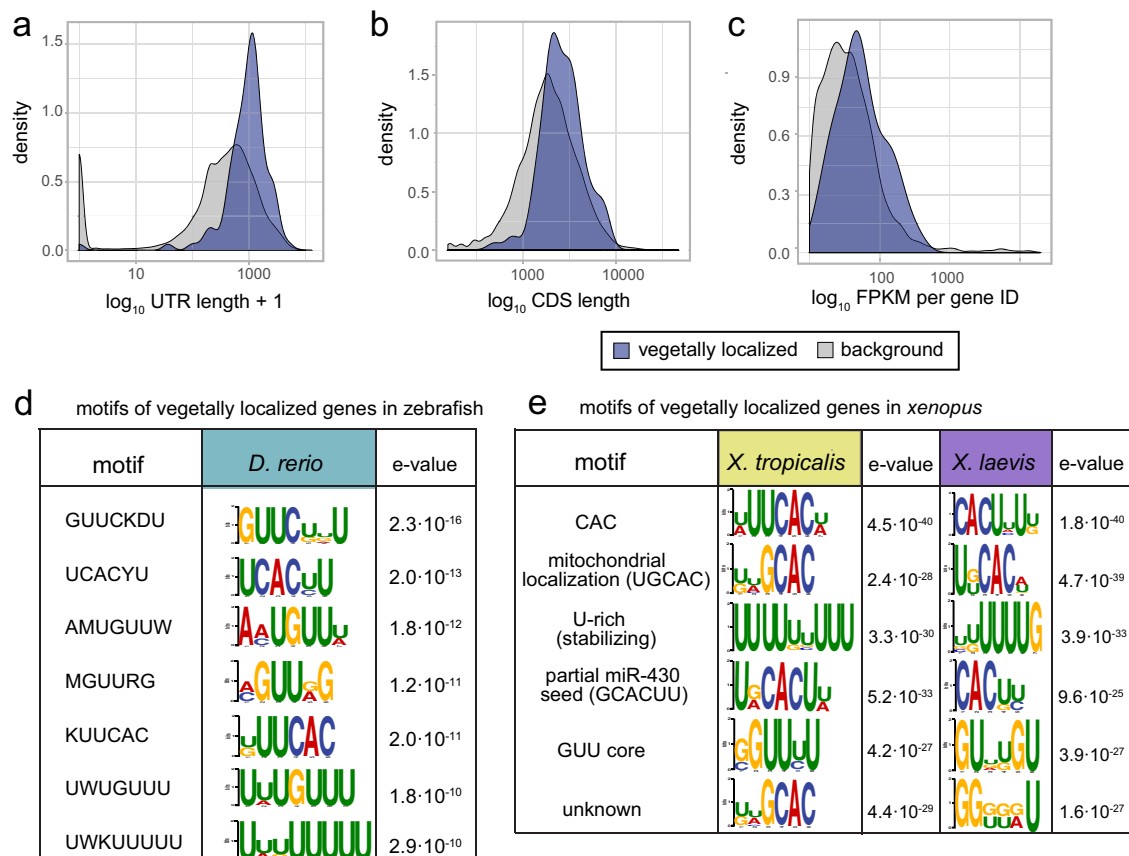

**Fig. 6 3′UTR characteristics of vegetally localized genes. a–c** Comparison of sequence characteristics of expressed isoforms of vegetally localized to all genes. **a** Weighted 3′UTR (untranslated region) lengths: isoforms contribute according to their relative expression, mean(vegetal genes) = 1.06 kb, mean (background) = 0.6 kb, $p$ value < $2.2 × 10^{-16}$ (two-sample Wilcoxon test). **b** Weighted lengths of coding sequences (CDS), mean(vegetal genes) = 2.78 kb, mean(background) = 2.42 kb, $p$ value = $7.655 × 10^{-14}$ (two-sample Wilcoxon test). **c** FPKM (fragments per kilobase of transcript per million mapped reads) sum per gene ID, IDs with <10 FPKM were omitted. Mean expression of vegetal genes 64.1 FPKM, mean of background 37.4, $p$ value < $2.2 × 10^{-16}$, (two-sample Wilcoxon test). **d** Results of the $k$-mer enrichment analysis of 3′UTRs of 216 expressed isoforms, zebrafish vegetally localized genes. Top seven motifs and logos. **e** Results of the $k$-mer enrichment analysis of the longest 3′UTR of vegetally localized genes in *X. laevis* and *X. tropicalis*, top six motifs, and their respective description based on previous publications.

would be needed to reconstruct the spatial transcriptome of a complete oocyte or fertilized egg.

Several methods have been published that pair scRNA-seq data with spatial imaging data, including two novel approaches using optimal transport theory[56,57]. This raises the intriguing possibility that it might be possible to integrate a limited number of in situ images with our tomo-seq data to predict spatial patterns transcriptome wide with higher resolution. However, one important difference of our sub-single-cell system compared to scRNA-seq is that we cannot directly use co-expression in single cells as a coupling between genes. Hence, other types of couplings, e.g., shared localization elements between genes, would be required.

We identified spatial patterns in the tomo-seq data based on SOM. However, it's important to note that detection of spatial expression patterns is a challenging computational task, and the performance of analysis methods will depend on the data type (e.g., 1D versus 2D methods) and on the type of spatial patterns (e.g., clearly delimited spatial domains versus smooth gradients).

For scSLAM-seq, we achieved two important advances: we made the method compatible with high-throughput scRNA-seq based on widely used droplet microfluidics approaches by performing the chemical derivatization of 4sU in intact methanol-fixed cells. Furthermore, we successfully established the method in early zebrafish embryos by labeling zygotically transcribed RNA via injection of 4sUTP into the zygote. This approach

allowed us to track the fate of individual maternal transcripts from the zygote until gastrulation. scSLAM-seq is a universal approach for following the fate of RNA molecules over time, and we anticipate that this strategy will emerge as a powerful method for short-term fate tracking of RNA molecules in living organisms. However, it is important to note that efficient delivery of 4sU into other live animals may require different approaches depending on the species and the organ system.

The combination of tomo-seq and single-cell RNA labeling generates important synergy by allowing transcriptome-wide spatio-temporal RNA measurement. We used this combination of techniques for a systems-level analysis of RNA dynamics in early zebrafish development, which gave us access to developmental events that are not captured by conventional scRNA-seq. Beyond this specific biological application, we anticipate that the combination of spatial transcriptomics and RNA labeling will find important applications for many other questions, such as tissue remodeling in disease conditions or analysis of cell–cell signaling interactions in vivo.

Besides the methodology presented here, another major output of this work consists in the transcriptome-wide resource of localized genes in three vertebrate species. While high-resolution atlases of transcript localization have been established in *Drosophila* oocytes based on automated microscopy[58], no comparable datasets exist for early vertebrate development, with the exception

of low-resolution spatial analysis of *Xenopus* oocytes[43–45]. Our analysis provides a shortlist of candidate genes with a potential role in early development, including genes like the phosphatase *ppp1r3b* or the kinase *camk2g1*, which have no known function in early embryogenesis, but are vegetally localized in all three species. We observed that many vegetally localized transcripts are later specifically transported into the PGCs, suggesting that specification of the PGCs is one of the main functions of the localized genes discovered here. We observed a relatively low conservation of localized genes, but a rather high conservation of enriched motifs in 3′UTRs. While it is possible that our analysis underestimates the true degree of conservation of vegetal localization due to the difficulty of reliably calling localization patterns for lowly expressed genes, this observation raises the question whether the function of genes involved in, e.g., PGC specification is conserved, even if the localization pattern is not.

Asymmetric localization of mRNA molecules is a pervasive phenomenon in the animal kingdom[59–61] and provides an important layer of gene regulation in a variety of different cell types by, e.g., restricting translation spatially[59,62] or by controlling translation efficiency[60]. While the exact nature of the localization motifs in early embryos have largely remained elusive, there are indications that secondary structure[63,64] or sequence-dependent piRNA adhesion traps might be involved[65]. While our high-resolution spatial transcriptomics data allowed a systematic analysis of enriched *k*-mers, the results probably do not reveal the full mechanism, since we did not identify a single motif that explains localization of all genes. This, together with the observation that 3′UTRs of vegetally localized genes are longer than for other genes, suggests more complex and potentially longer regulatory elements than the *k*-mers analyzed here. We speculate that the combination of tomo-seq with the injection of 3′UTR fragments may in the future provide further insights into the molecular mechanisms underlying RNA localization.

## Methods
### Animal methods
*Breeding of zebrafish.* Fish were maintained according to standard laboratory conditions. All animal procedures were conducted as approved by the local authorities (LAGeSo, Berlin, Germany), and we complied with all relevant ethical regulations for animal testing and research. For embryo experiments, we set up group crosses of AB wild-type fish originally obtained from Karlsruhe Institute of Technology (KIT), European Zebrafish Resource Center (EZRC; catalog number #1175).

*Preparation of frog oocytes.* Oocyte from wild-type animals were ordered from the European *Xenopus* Resource Centre at the University of Portsmouth (https://xenopusresource.org) as dissected ovary lobes on wet ice. Oocytes were manually dissected with forceps on agarose plates, and gently dissociated with liberase as described by Claussen et al.[66].

### Laboratory methods
*Tomo-seq.* Zebrafish embryos were harvested 20 min after fertilization. Individual embryos were embedded in OCT medium under a dissection microscope and oriented along the animal–vegetal axis with tungsten needles. Since the transparency of the embryo makes the embryo invisible after freezing the block, we marked the starting point for the blind collection of sections with a blue polyacrylamide bead (BIORAD). Before snap-freezing the cryomold on dry ice, we took a picture to calculate the distance between the edge of the block and the polyacrylamide bead in Fiji. The time point of snap-freezing corresponded to ~30 min post fertilization. While it would be interesting to analyze earlier stages (i.e., 5 min after fertilization), this might entail more technical variation due to the time needed for precise embedding.

We sectioned the blocks into 96 sections (thickness 10 μm), added 1 μl ERCC spike-in controls (diluted 1:50,000) and 0.5 μl Glycoblue; and extracted RNA with Trizol as described in Holler and Junker [15]. Pelleted and dried RNA was directly dissolved in a mix of dNTPs and barcoded poly-dT primers, and was reverse-transcribed with SuperScript II. Primer design was inspired by CELseq2 (ref. [3]), using 8 nt barcodes, 6 nt UMIs, and a modified adapter design (see Supplementary Dataset 8 and https://github.com/karolineholler/tomo-seq). The following steps include second strand synthesis, linear amplification with IVT, RNA

fragmentation, second reverse transcription with SuperScript III and a library PCR. A detailed protocol can be found in Holler and Junker[15]. For *Xenopus* oocytes, we used the same protocol, but adjusted the section thickness according to the sample diameter and the RNA fragmentation time to a higher RNA input.

*Bulk RNA sequencing.* Embryos were harvested 20 min after fertilization and directly put into Trizol. We extracted RNA with chloroform and isopropanol, and dissolved the pelleted RNA in nuclease-free water. Quality of the RNA was checked on a bioanalyzer RNA pico chip. We then prepared full-length sequencing libraries with the Illumina TruSeq stranded mRNA kit. The samples were sequenced on Illumina HiSeq4000.

*Whole-mount in situ hybridization.* Zebrafish embryos were fixed 20 min after fertilization in 4% PFA for 2 h. Whole-mount in situ hybridization was performed as in Thisse et al.[67].

*scSLAM-seq*

### 4sUTP injections
We injected zebrafish embryos directly after fertilization with 4 nl 4sUTP (12.5 mM, Sigma-Aldrich, in 10 mM Tris·HCl pH 7.4, Carl Roth). At 50% epiboly, we removed the chorions, then continued incubation until shield stage.

### Cell fixation and iodoacetamide treatment
We dissociated ten shield stage embryos per sample by gently pipetting up and down in deyolking buffer (55 mM NaCl, 1.8 mM KCl, 1.25 mM NaHCO₃ in HBSS, Life Technologies). For cell fixation, we added cold methanol (Carl Roth) until a final concentration of 80%. We then fixed the cells at −20 °C for 30 min. For chemoconversion, we added 1 M IAA (Sigma-Aldrich) in 80% methanol and 20% HBSS to a final concentration of 10 mM, and gently agitated the mixture at room temperature, overnight, in the dark.

### Rehydration and preparation for scRNA-seq
To inactivate the IAA, we spun down the cells at $1000 \times g$ for 5 min and resuspended in quenching buffer (DBPS, Gibco, 0.1% BSA, Sigma-Aldrich, 1 U/μl RNaseOUT, Life Technologies, 100 mM DTT, Carl Roth) and incubated at room temperature for 5 min. After spinning down again, we resuspended them in DPBS containing 0.01% BSA, 0.5 U/μl RNaseOUT and 1 mM DTT. The cells were then passed through a 35 μm strainer, counted, and immediately loaded onto a 10x Chromium system using the 3′ kit (V2 and V3).

### Library preparation and sequencing
We prepared sequencing libraries according to the manufacturer's instructions and sequenced them on Illumina HiSeq4000 and NextSeq500 systems.

### Dot blots for detection of incorporation and IAA derivatization of 4sUTP
We biotinylated extracted RNA using the following mixture: 70 ng RNA in 96.8 μl water, 2 μl 1 M Tris•HCl (pH 7.4, Carl Roth), 0.2 μl 0.5 M EDTA (Carl Roth) and 1 μl 10 mg/ml MTSEA-XX-Biotin (Biotium). The reaction was incubated at room temperature for 30 to 60 min in the dark. We then separated the biotinylated RNA from excess biotin by adding the same volume of phenol:chloroform:isoamylalkohol (Sigma-Aldrich), mixing well and spinning in Phase-Lock-Gel tubes (Quantabio) at $15,000 \times g$ for 5 min. The RNA was then transferred on a Hyperbond N+ membrane (Amersham) and UV crosslinked with 2400 μJ (254 nm). To block non-specific signal, we incubated the membrane in blocking solution (PBS pH 7.5 (Gibco), 10% SDS (Roti®-Stock 20 % SDS, Carl Roth), 1 mM EDTA) for 30 min. The membrane was then probed with a 1:5000 dilution of 1 mg/ml streptavidin–horseradish peroxidase (Pierce) in blocking solution for 15 min. Finally, the membrane was washed six times in PBS containing decreasing concentrations of SDS (10, 1, and 0.1% SDS, applied twice each) for 10 min. The signal of biotin-bound HRP was visualized using Amersham ECL Western Blotting Detection Reagent (GE Healthcare).

Flp-In™ 293 cells (Thermo Fisher; R75007) used as a positive and negative control were grown in DMEM (Gibco) + 10% FBS (Gibco) + 2 mM L-glutamine (Gibco) at 37 °C and 5% CO₂. The cells were incubated with 300 μM 4sU or mock treated for 15 min before we fixed them in methanol, as described above.

### Quantification and statistical analysis
*Mapping of tomo-seq data.* Fastq files were mapped with STAR (v2.5.3a) using the -quantMode option. Genome versions used were GRCz10 (*Danio rerio*), 9.2 (*X. laevis*) and 9.1 (*X. tropicalis*). From the SAM file, gene counts were assigned to a spatial barcode resulting in a count matrix. A mapping script can be accessed via github: https://github.com/karolineholler/tomo-seq[68].

*Further processing of tomo-seq data.* Data analysis was performed in R (v3.6.0) using custom code. We filtered out sections with a low recovery of ERCC spike-in controls. The cutoff depends on the sequencing depth, and was set as ~0.04 percent of the mapped reads of a library (or 8000 transcripts for the replicate shown in Fig. 1). In the remaining sections, we excluded lowly expressed genes (with less than five counts in at least one section, for the replicate shown in Fig. 1), then divided gene counts by total counts in that section and normalized to the median section size. For clustering based on SOM, we calculated cumulative expression going from low to high section numbers, normalized the maximum of the cumulative expression to one and let the SOM sort these patterns into a linear matrix of $1 \times 50$ profiles. A gene was called vegetally localized in all replicates when it was assigned any profile between 46 and 50 in all replicates and at least 48 in one replicate. Code for data analysis can be found at github: https://github.com/karolineholler/tomo-seq[68].

*Isoform analysis and k-mer enrichment.* Isoform expression in zebrafish one-cell stage embryos was determined using cufflinks v2.2.1. For *k*-mer enrichment, we extracted 3′UTR sequences as annotated in the zebrafish genome version GRCz10. Next, we compared vegetally localized to all expressed genes with DREME (v4.11.2) using the parameters: -g 1000 -norc -e 0.5 -mink 3 -maxk 10. For *Xenopus*, we used the longest annotated 3′UTR for our analysis. We calculated the 3′UTR length of a gene ID as shown in Fig. 6a by weighing the isoforms 3′UTR length, according to their relative contribution to a gene IDs total expression. CDS length as shown in Fig. 6c were calculated accordingly.

*Alignment of UTRs from D. rerio, X. laevis and* X. tropicalis. UTR sequences were aligned using the mafft online tool (http://mafft.cbrb.jp/alignment/server/), using the following parameters: %mafft -reorder -anysymbol -maxiterate 1000 -retree 1 -genafpair input.

*scSLAM-seq mapping and analysis.* Raw data were demultiplexed with cellranger mkfastq (v3.0.2), and mapped with the default parameters of cellranger (10× Genomics) to the zebrafish genome, version GRCz11.95. We used the inbuilt cell detection algorithm to create a "whitelist" with all barcodes that contain cells and extracted these barcodes from the BAM file to only consist of reads from real cells. We further separated the reads in that file into labeled reads (>1 T to C mutation per UMI, base quality >20) and unlabeled reads. We then created a fastq file for labeled and for unlabeled reads, respectively, mapped them with STARsolo and obtained count matrices that were further analyzed with Seurat v.3.1.2. The code for mapping and data analysis is publicly available via https://github.com/karolineholler/scSLAM-seq[69].

*Analysis of ligand–receptor interactions.* We used CellPhoneDB[70] and CellChat[71] to identify maternal genes involved in potential ligand–receptor interactions. Human gene names were converted to zebrafish genes using orthology data from the Alliance of Genome Resources, release 3.2.0 (ref. [72]). We added up contributions from zebrafish genes that have the same human orthologue, and we removed orthologue pairs where one zebrafish gene can be converted to multiple human genes. We then calculated the fraction of maternal reads for the genes involved in interactions based on the raw count matrices for each cell type. We included only cells that had at least 150 labeled and at least 150 unlabeled features, resulting in a total of 6844 cells.

*Calculation of false negative rate in scSLAM-seq.* We estimated the false negative rate (i.e., the probability of a zygotic transcript molecule to remain unlabeled) with the following back-of-the-envelope calculation: we expect that ~5% of all Us are labeled in a zygotic transcript (Fig. 3c). The read length is 99 nt. Since the library was sequenced with ~4 reads per UMI, we assume an effective read length of 300 nt, taking into account that different reads for the same UMI may partially overlap. The GC content is on average 40%, which results in 30% Us, and hence 90 Us per transcript molecule. The probability that a zygotic transcript does not contain a single labeled U is therefore $0.95^{90} \approx 1\%$.

**Reporting summary**. Further information on research design is available in the Nature Research Reporting Summary linked to this article.

## Data availability
Raw data and count tables for tomo-seq and scSLAM-seq can be accessed on GEO under accession number "GSE158849". All other relevant data supporting the key findings of this study are available within the article and its Supplementary Information files or from the corresponding author upon reasonable request. A reporting summary for this article is available as a Supplementary Information file.

## Code availability
All scripts for mapping and analysis of scSLAM-seq[69] and tomo-seq[68] data are accessible via github (https://github.com/karolineholler/).

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

## Acknowledgements

The authors thank Jana Richter for her work on whole-mount in situ hybridizations, Ronny Schäfer for zebrafish injections, and Nora Fresmann for her kind support with scSLAM-seq experiments during the Sars-Cov2 lockdown of the laboratory. We thank Moritz Ophaus who produced *X. laevis* tomo-seq libraries as a rotation student in the lab, and Roberto Moreno Ayala for support with cell type identification of single-cell data. We also acknowledge support by MDC/BIMSB core facilities (zebrafish, genomics, bioinformatics). Work in J.P.J.'s lab was supported by a European Research Council Starting Grant (ERC-StG 715361 SPACEVAR) and a Helmholtz Incubator grant (Sparse2Big ZT-I-0007).

## Author contributions

K.H. performed zebrafish tomo-seq experiments and data analysis. Tomo-seq of *X. tropicalis*, analysis and comparison to *X. laevis* data was done by J.M. as part of her master thesis under supervision of K.H. and J.P.J. *K*-mer enrichment analysis was done by P.B. and U.O. B.S. built the mapping pipeline, adapted self-organizing maps to tomo-seq data and assisted with statistical data analysis. Experimental method development of scSLAM-seq was done by A.N., R.A. and M.L., and A.N. developed the computational pipeline for scSLAM-seq analysis. Single-cell data were jointly analyzed by K.H. and A.N., PGC specific and statistical analyses were performed by K.H. and B.S. All authors discussed and interpreted the results. The paper was written by K.H. and J.P.J.

## Funding

## Competing interests

The authors declare no competing interests.
