## [Peer Review File · Nature Communications]

Reviewers' Comments:

Reviewer #1:

Remarks to the Author:

In this manuscript the authors improved and combined two methods, the tomo-seq method for spatially-resolved transcriptomics and a single-cell metabolic RNA-seq labeling method, to analyze mRNA spatiotemporal dynamics in zebrafish embryo. The overall combined approach is potentially novel in tracking mRNA dynamics from few cells to many cells during early embryo development. The biological questions studied here are significant and interesting. However, several important points need to be clarified or addressed before publication. Below is a list of specific questions and concerns.

- During the first few divisions of the egg into 2, 4, 8 and 16 cells all of the cells (blastomeres) are in cytoplasmic continuity with the yolk - there is a partial plasma membrane separating them from each other but potentially mixing of gene products through their common connection with the yolk cell. In addition, all of these divisions and all cells of the embryo up until there are several thousand, lie at the animal pole, while the vegetal pole remains one big yolk cell. It is only after 6 hours that the animal pole cells spread down to the vegetal pole to surround the yolk. That is fundamentally different from the frog embryo, in which the yolk does subdivide into vegetal blastomeres. To some extent it is difficult to relate animal-vegetal localization in the egg with cells at later stages. Is a single-nuclei seq approach needed?
- There are three major results in the paper: a) improvement of the two methods, b) labeling maternal transcripts and tracking them, c) applications of the tool to three developmental systems. The Results section was written without subtitles and the overall messages were difficult to follow. The part b) seems to be the most novel component of the method. The Results clearly benefit from better organization with subtitles, and the part b) could be emphasized more.
- The current study has improved the original tomo-seq (2014) developed by the authors for better spatial resolution. However, sequencing 1-d slice of cells using tomo-seq still has a limitation on distinguishing some patterns, e.g. radial geometry. For genes that are localized in a local compact region, examining the sample with different directional slices, for example, vertical and horizontal, could potentially improve the measurement. Compared to other 2D methods (e.g. SeqFISH+), what is the major advantage of the new improved tomo-seq method?
- It is clear that adding some temporal trajectory analysis (e.g. pseudotime or lineage relationship) and gene-gene network analysis of the measured single-cells collected at several time points can provide new insights on how maternal genes and other genes interplay for spatial patterning during early development.
- While both zebrafish and frog embryos are analyzed in the study, it is unclear what new and significant biological findings are besides the general point of evolutionary conservation and confirmation of several known findings.
- In Figure 2a, the spatial pattern of genes is sorted by SOM. With the recent development of spatial transcriptomics, there have been methods introduced specifically for identifying spatial patterns, such as SpatialDE. Using the specialized methods might provide better analysis for the patterns.
- The scales of the two axes look quite different in Figure 1e. Why?

Reviewer #2:

Remarks to the Author:

Manuscript by Holler et al describes a new resource. Authors improved a previously established tomo-seq method to measure RNA localization in 1-cell stage zebrafish embryos. They also developed an approach based on 4su metabolic labeling of newly synthesized transcripts in zebrafish embryos and performed single-cell RNA-seq which allowed to predict enrichment of

vegetally or animal pole derived transcripts to different cell types. And finally, the authors applied tomo-seq approach to study localization of maternal transcripts in *Xenopus tropicalis* and *Xenopus laevis* species. Motif analysis of 3'UTR sequences identified candidate sites overrepresented in vegetally localized genes.

Overall, the study is well performed, the statistical analysis and conclusions are appropriate, the results are novel and will provide a useful resource for mRNA localization and functional studies. Tomo-seq approach by itself is not novel, and has been used previously. However, using higher resolution and improved analysis approaches, Holler et al identified ten-fold higher number of vegetally localized mRNAs than previously known. Application of 4sUTP labeling for mRNA labeling in vertebrate embryos is novel and results in unique information. However, the major conclusion of this experiment is that vegetally localized genes are enriched in PGCs, which by itself is not particularly novel and has been previously demonstrated using conventional approaches. Evolutionary studies of vegetal mRNAs and 3'UTR element characterization is intriguing. However, it is not clear if any of identified candidate sequences have a functional significance, and this study would be greatly enhanced by functional characterization of some of these 3'UTR elements.

Specific points:

1. Supplementary tables would benefit from a brief legend included at the title line which explains annotations in the table.
2. The list of 97 vegetally localized genes should be provided in a separate table, which also includes annotations of which genes have been previously shown to be localized vegetally. A separate table of genes localized to the animal pole should be also provided. It is difficult to extract this data from Supplemental Table 1 in its current form.
3. The manuscript argues that 47 vegetally localized genes were significantly enriched in PGCs, and 28 of them were marker genes for PGCs. It is difficult to see this result in Suppl. Table 3 which has marker genes for all clusters listed. The authors should provide a separate table that lists all 47 genes and note fold change in different cell types, and also note which 28 genes were among marker genes for PGCs.
Also, please check Suppl. Table 3; there are duplicate cluster assignments listed for a subset of genes (*cdx4*, *ved*, *hes6* and others)
4. The previous studies noted either very few (Owens et al) or a large majority of genes (Sindelka et al) enriched at the animal pole in *Xenopus* embryos. How did the list of animally localized genes identified in the current study compare with the genes identified in the previous studies? It would be helpful to briefly discuss potential reasons for the differences between studies.
5. Authors found low evolutionary conservation between vegetally localized genes. A trivial possibility could be that many of these genes showed low expression in some of the species and were filtered out during the analysis. What was the localization of the 97 zebrafish vegetal genes in other species? Did these genes show different localization pattern, or were they just not detected during analysis in other species?
6. The authors describe 9 conserved vegetally localized genes. What was the localization of each of these 9 genes at gastrula stage based on scSLAM-seq analysis?
7. What was the localization of *anln* in *X. laevis*? The text says that it was not vegetally localized but it is not clear what its localization pattern was.
8. Analysis of 3'UTR identified candidate motives enriched in vegetally localized genes. The manuscript would greatly benefit from experimental validation of these motives. Are any of them required or sufficient for vegetal localization? Injection of a reporter mRNA which has these sequences intact or mutagenized could be used to answer this question.

Reviewer #3:

Remarks to the Author:

Summary

Holler et al investigated the spatial localization of mRNAs in the single-cell zebrafish embryo and distinguished maternal and zygotic transcripts at gastrulation stage using recently developed sequencing technologies. Using these data, they find that vegetally localized genes in the one cell embryo are enriched in primordial germ cells at gastrulation. Based on previous literature on conservation of 3'UTR elements that may drive transcript localization across species, they asked whether localized genes found in zebrafish are shared in *Xenopus laevis* and *tropicalis* embryos. They identified 9 genes that localize vegetally in zebrafish and two species of *Xenopus* and further

investigated sequence elements in 3'UTR of these genes to find repeated motifs that are linked to characterized functions such as RNA stability and motifs not yet described.

This manuscript provides datasets that should be useful for others in the field interested in maternal mRNA localization and turnover, it showcases two relatively new sequencing technologies, and provides some new biological insight into germ cell localized RNA, so I support publication of this manuscript. My comments are aimed mostly at helping the authors properly contextualize and describe their findings.

Text

After reading the abstract (and indeed the introduction), it is not exactly clear what the authors did experimentally. As this is largely a technique driven paper, I think it is useful to spend a sentence naming and/or describing the tomo-seq and scSLAMseq technique in the abstract and intro. This is especially important as there are new versions and competing approaches constantly being developed, so it would be nice for readers to know early which techniques these were.

Since there were two timepoints captured - one-cell stage and gastrulation, "dynamics" (including the title) might not be accurate.

Introduction could use some reframing to tie in the biological question investigated (RNA transport in early development?) and then lead into the limitations of tools available and how the authors utilized combinations of approaches and developed new ways to interrogate this question.

In results, it would be useful to better justify the orientation and timing chosen for tomo-seq with reference to what is known about vegetally localized mRNAs and microtubule transport. The equivalent of cortical rotation in zebrafish has already happened at 30 minutes, so this is a bit of a late stage to choose for tomo-seq with an animal-vegetal orientation. There is an argument that an earlier stage (5 minutes after fertilization, i.e. before cortical rotation) should have been chosen to identify vegetally localized mRNAs in the egg using tomo-seq along the animal-vegetal axis and then dorsal-ventral tomo-seq at 30 minutes post fertilization.

The authors repeatedly say "sub single cell tomo-seq". While technically true, the fertilized egg is a very unique cell and tomo-seq in general is not a good method for sub single cell transcriptomics compared to FISH based approaches.

[lines 177-179] To suggest that *wnt8a* and *syntabulin* are degraded more rapidly than germ cell factors, further support from literature and/or validation needed, or an argument that their method has the dynamic range to detect such a lowly expressed gene.

The authors should add p-values to the k-mer enrichment analysis of 3'UTRs for localized genes. The identified motifs are short and unclear if they are enriched by chance. Especially, since they are not experimentally tested.

Citations

Should cite classic papers on *vg1* mRNA localization in *Xenopus* from Melton lab.

[line 91] That the transcripts that are enriched in the vegetal pole are "specifically transported and retained" needs citation.

[line 104 and 105] Citation needed to support "..., which increases the number of known vegetal genes by about tenfold" and "...have previously been shown to localize vegetally."

[Figure 5a ii] Citation needed for the phylogenetic tree and label figure with species names.

Format/ minor text changes

[Figure 2d] Axis labels missing and scale bars missing on images of embryos. Panel could be larger.

[Figure 5a iii] Label what the black dots represent or note in figure legend. Not sure whether this panel is necessary?

For the supplementary tables, could add captions for what each column represents.

Summary

We would like to thank all reviewers for the interest they expressed in our manuscript as well as for their constructive criticism, which allowed us to improve the manuscript. We discuss these changes in detail in the attached pages, but also highlight selected key points here:

- We now highlighted the advantages of our experimental approach more clearly, and we clarified our new biological findings.
- We now expanded our scSLAM-seq analysis from vegetally localized genes to all maternal genes, including a new unbiased ligand-receptor analysis in Fig. S5.
- We compared our computational approach for detection of spatial patterns to the SpatialDE method. We find that, in our dataset, our SOM approach allows identification of more vegetal genes than SpatialDE.
- We expanded the supplementary tables and provided better annotations of the tables, in order to make it easier for other researchers to use our data.
- We now performed an in-depth comparison of our xenopus data to previous publications, and we investigated possible technical reasons for the low conservation of vegetally localized genes between the three species.
- We now performed mRNA injections of reporter constructs into cultured immature zebrafish oocytes. This is not an established approach in the field, so our experiments should be considered as pilot experiments.
- To increase clarity, we revised the Abstract and Introduction sections, changed the title of the manuscript, and added statistics and references where necessary.

Reviewer #1 (Remarks to the Author):

In this manuscript the authors improved and combined two methods, the tomo-seq method for spatially-resolved transcriptomics and a single-cell metabolic RNA-seq labeling method, to analyze mRNA spatiotemporal dynamics in zebrafish embryo. The overall combined approach is potentially novel in tracking mRNA dynamics from few cells to many cells during early embryo development. The biological questions studied here are significant and interesting. However, several important points need to be clarified or addressed before publication. Below is a list of specific questions and concerns.

We would like to thank the reviewer for the positive assessment of our work and for the constructive feedback.

1) During the first few divisions of the egg into 2, 4, 8 and 16 cells all of the cells (blastomeres) are in cytoplasmic continuity with the yolk - there is a partial plasma membrane separating them from each other but potentially mixing of gene products through their common connection with the yolk cell. In addition, all of these divisions and all cells of the embryo up until there are several thousand, lie at the animal pole, while the vegetal pole remains one big yolk cell. It is only after 6 hours that the animal pole cells spread down to the vegetal pole to surround the yolk. That is fundamentally different from the frog embryo, in which the yolk does subdivide into vegetal blastomeres. To some extent it is difficult to relate animal-vegetal localization in the egg with cells at later stages. Is a single-nuclei seq approach needed?

We agree with the reviewer that a single-nuclei-seq approach would be interesting for analyzing differences between individual cells at blastomere stages, since the separation of the cells is incomplete. However, even a single-nuclei-seq experiment would provide only limited insight, since maternal transcripts will probably be largely excluded from nuclei. Due to this limitation we focused our analysis on the vegetal pole at the one-cell stage, and performed scRNA-seq only at later stages.

As the reviewer points out, the structure of the early zebrafish and the early frog embryo are fundamentally different – one big yolk cell in zebrafish, and subdivision of the yolk at cell division in xenopus (illustrated in Fig. 5a (iii)). In the revised manuscript we now describe this difference more clearly. In particular, we now discuss that the difficulty in relating animal-vegetal localization in the egg to cells at later stages motivated us to establish scSLAM-seq in the early zebrafish embryo.

2) There are three major results in the paper: a) improvement of the two methods, b) labeling maternal transcripts and tracking them, c) applications of the tool to three developmental systems. The Results section was written without subtitles and the overall messages were difficult to follow. The part b) seems to be the most novel component of the method. The Results clearly benefit from better organization with subtitles, and the part b) could be emphasized more.

We thank the reviewer for this comment. We have now added subheadings in the Results section to increase clarity, and we highlighted the labeling and tracking of maternal transcripts more strongly by changing the title and the Discussion.

3) The current study has improved the original tomo-seq (2014) developed by the authors for better spatial resolution. However, sequencing 1-d slice of cells using tomo-seq still has a limitation on distinguishing some patterns, e.g. radial geometry. For genes that are localized in a

local compact region, examining the sample with different directional slices, for example, vertical and horizontal, could potentially improve the measurement. Compared to other 2D methods (e.g. SeqFISH+), what is the major advantage of the new improved tomo-seq method?

We agree with the reviewer that 1D slicing limits the resolution for certain spatial patterns. As suggested by the reviewer, 3D reconstruction based on incorporation of different slicing directions could alleviate this issue. However, due to the external radial symmetry of the zebrafish one-cell stage embryo, the additional slicing directions would be somewhat ill-defined, unless markers for the (future) DV axis are introduced. That said, we agree with the reviewer that tomo-seq along an axis orthogonal to the animal-vegetal axis might reveal additional details regarding the spatial organization of the transcriptome. In the revised manuscript we discuss the resolution limits of our 1D tomo-seq approach, and we now explicitly state that our current approach cannot reveal the full spatial organization of the transcriptome in the one-cell stage embryo. Furthermore, we now discuss our methodology in comparison to approaches for spatial transcriptomics in 2D. Methods like SeqFISH+ have not been successfully applied to oocytes or fertilized eggs yet, and the high autofluorescence of the yolk might lead to experimental challenges for such microscopy-based approaches. Methods like SeqFISH certainly hold great promise for precise spatial transcriptomics. However, it is important to note that analysis of a large number of serial sections would be required to cover the complete oocyte or fertilized egg, rendering the approach somewhat impractical, since sequential hybridization is a time-consuming process already for a single section.

We have added the following paragraph in the Discussion:

“The tomo-seq method is well suited for spatial transcriptomics in one-cell stage zebrafish embryos, since we expect the most striking patterns along the animal-vegetal axis. However, more complex spatial patterns, including e.g. radial geometry, would be difficult to detect with our approach, which is based on serial sections in 1D. Different strategies would be required to reveal the full spatial organization of the transcriptome in 3D. While sequencing-based approaches for spatially-resolved transcriptomics in tissue sections typically do not reach the spatial resolution required here (Rodrigues et al., Science, 2019) methods based on sequential fluorescent in-situ hybridization (Eng et al., Nature, 2019) have the potential to reveal more complex spatial patterns than can be detected by tomo-seq. However, analysis of a large number of serial sections would be required to reconstruct the spatial transcriptome of a complete oocyte or fertilized egg.”

4) It is clear that adding some temporal trajectory analysis (e.g. pseudotime or lineage relationship) and gene-gene network analysis of the measured single-cells collected at several time points can provide new insights on how maternal genes and other genes interplay for spatial patterning during early development.

We would like thank the reviewer for the suggestion to further investigate the potential role of maternal transcripts in developmental cell fate decisions. In the submitted version of the manuscript we had focused the analysis of the scSLAM-seq data on the vegetally localized genes. Inspired by the reviewer’s comment we now explored potential interactions of maternally deposited genes more broadly. The reviewer suggests two interesting analytical approaches, interaction analysis and pseudotemporal ordering. We addressed the two parts of the reviewer’s question separately, as described below:

Interaction analysis

Interaction analysis can be performed on two levels: Reconstruction of gene regulatory networks in cells, or analysis of ligand-receptor interactions between different cells. Single-cell multi-omics data holds great promise for reconstruction of gene regulatory networks. However, current methods for identification of gene regulatory networks have only moderate performance if based solely on scRNA-seq (Pratapa et al., Nature Methods, 2020). We therefore decided to focus our attention on cell-cell interactions. Using CellPhoneDB (Efremova et al., Nature Protoc, 2020) and published Genome orthology data (Alliance of Genome Resources Consortium, Nucleic Acids Res, 2020), we identified ligand-receptor pairs between cell types, which correspond to potential signalling interactions. The statistically significant ligand-receptor pairs are displayed below. Fig. R1-1 shows the maternal mRNA fraction of the interaction partners in the respective cell types. There is a clear enrichment of maternal ligands and receptors in PGCs, while the potential interaction partners are in a variety of cell types. In Fig. R1-2 we display the genes that underlie these interactions. In this plot, we computed a “maternal score” for each of the ligand-receptor interactions. This score is calculated as the average of the fractions of maternal transcripts for the ligand and the receptor in their respective cell types (0 means that the interaction is driven exclusively by zygotic transcripts, and 1 means it is driven exclusively by maternal transcripts). *Lamp1* stands out as a PGC-enriched maternal gene that has to potential to interact with other cell types via the secreted ligand *Fam3c*. We now included this analysis as a new supplemental figure (Fig. S5b and c).

Fig. R1-1. Maternal mRNA fraction of the potential interaction partners identified by ligand receptor-analysis.

Fig. R1-2. Gene pairs involved in the potential cell-cell interactions shown in Fig. R1-1. Color code corresponds to the “maternal score” of the respective ligand-receptor interaction.

Pseudotemporal ordering

Two major previous publications used time series scRNA-seq data to reconstruct differentiation trajectories in early zebrafish development (Wagner et al., Science, 2018; Farrell et al., Science,

2018). These publications show that the timepoint of our dataset (6 hpf) only covers the very first steps of trajectory branching (see Fig. 2b in Wagner et al.), which highlights the difficulties of trajectory analysis at these early stages. The UMAP representation of our data in Fig. 3d shows a trajectory from undifferentiated cells to enveloping layer cells. Since the enveloping layer is characterized by particularly low levels of maternal transcripts, this trajectory is not particularly suitable for analyzing the role of maternal transcripts in cell differentiation. PGCs, on the other hand, are characterized by high levels of maternal transcripts, and are known to be specified by maternally deposited germ plasm. However, we do not detect a differentiation trajectory towards PGCs in Fig. 3d. Therefore, the RNA tracking approach we had taken in Fig. 4 appears to be a more productive strategy for identifying maternal transcripts involved in PGC formation compared to pseudotemporal ordering. All other cell types are still at very early stages of differentiation, which would render a trajectory analysis unreliable. However, to address the reviewer's question, we now expanded the cell type enrichment analysis from Fig. 4a from vegetally localized genes to all maternal transcripts. We observed that the majority of maternal transcripts are not enriched in any specific cell type, but for a small fraction of genes we found an enrichment of maternal transcripts in PGCs, and to a lesser degree also in enveloping layer cells (Fig. R1-3). This analysis is now included in Fig. S5a in the revised manuscript.

Figure R1-3. Fold change enrichment of maternal transcripts for different cell types vs. all other cells. Genes with an average expression lower than 0.1 transcripts/cell were excluded from this analysis.

5) While both zebrafish and frog embryos are analyzed in the study, it is unclear what new and significant biological findings are besides the general point of evolutionary conservation and confirmation of several known findings.

As the reviewer correctly points out, our comparison between zebrafish and xenopus (Fig. 5 and 6) confirms several known findings (e.g. similar localization motifs, but limited overlap of

localized genes). Since establishing new experimental and computational approaches is an important part of our manuscript, we believe the confirmation of known findings is a valuable validation of our approach.

Furthermore, for the following reasons we believe that our species comparison may also serve as a valuable resource for future research: 1) Identification of conserved vegetally localized genes provides a shortlist of interesting candidate genes for functional analysis. 2) While previous annotations of animally localized genes in xenopus were rather questionable (containing either very few or a majority of all genes), we now provide a distinct list of animal genes. 3) Identification of conserved sequence elements in 3'UTRs of vegetally localized genes opens the door to functional experiments.

Regarding the last point, the reviewer's comment inspired us to attempt a functional experiment in which we aimed to test the effect of conserved enriched motifs by injection of fluorescently labeled reporter constructs into localization-competent oocytes. For analysis of mRNA localization, injection into immature oocytes is required, since the localization process is already completed in mature oocytes, and the localization machinery is not active any more in fertilized eggs (Pelegri, *Dev Dyn*, 2003, Kosaka et al., *Mech Dev*, 2007). Since injection and culturing of immature zebrafish oocytes is not a standard technique, we performed the following series of pilot experiments:

As a first experiment, we cultured extracted oocytes for 22h without injection (Nair et al, *Dev Dyn*, 2013). Live/dead staining (FDA and PI) revealed a mixture of healthy and dying cells (Fig. R1-4)

Figure R1-4: Live and dead stain of extracted zebrafish oocytes.

Despite attempts at optimizing the extraction procedure and the culturing conditions, we were unable to obtain a larger fraction of live cells. In a next set of experiments, we injected a reporter construct consisting of the CDS of dTomato and the 3'UTR of the well-characterized vegetally localized gene *dazl* as a positive control (Figure R1-5 a). The mRNA was fluorescently labeled with Aminoallyl-UTP-ATTO488 to allow direct detection of localization patterns by stereofluorescence microscopy. We injected the construct into oocytes of various sizes in order to cover oocytes at different stages of maturation. Immediately after injection, the mRNA could be detected as a cloud emanating from the point of injection (Figure R1-5 b). However, after culturing for 4h and 22h, the fluorescent signal had disappeared from most cells, suggesting that the mRNA had been degraded (Figure R1-5 c,d).

Figure R1-5: Injection of fluorescently labeled mRNA into zebrafish oocytes.

Of note, we detected strong red fluorescence in some cells (Figure R1-5 d), indicating that the injected mRNA was translated and persisted long enough to produce considerable levels of dTomato protein. In those cells with detectable remaining reporter mRNA, we did not observe a clear localization of the green signal at either 4h or 22h. At this point we decided to stop these experiments, since the most likely reason for the failure of the experiment is suboptimal oocyte culturing conditions, optimization of which is outside our field of expertise and might easily develop into a time-consuming project of its own. However, we believe that this is an interesting research question that would be suitable for a follow-up project.

6) In Figure 2a, the spatial pattern of genes is sorted by SOM. With the recent development of spatial transcriptomics, there have been methods introduced specifically for identifying spatial patterns, such as SpatialDE. Using the specialized methods might provide better analysis for the patterns.

Following the reviewer's suggestion, we now performed an analysis using SpatialDE to identify spatial patterns in our data. SpatialDE identified only 31 (out of 6863) genes as spatially differential in our dataset, compared to the 97 vegetal genes identified by our SOM analysis. There are important differences between SpatialDE and SOM: SpatialDE performs a significance test, which our SOM analysis does not include. Significance testing is particularly important for medical samples, where biological replicates are often not available or not possible. In our manuscript, we assessed the reproducibility of our results via replicate experiments (Fig. S2) and by in situ (Fig. 2).

In our previous analysis, we used cumulative normalized expressions for SOM to smooth over section-to-section (technical) variability and to make the resulting patterns more biologically informative. To our knowledge, SpatialDE does not do this, and it's possible that this is at least

partially responsible for the differences in performance. Importantly, it might be quite complex to implement this sort of cumulative smoothing in SpatialDE. While the null hypothesis could still be formulated as normal distributions for each section, the mean and standard deviation for each section then depend on those of the previous sections. Fitting and statistical testing with this model would require extensive rewriting of the SpatialDE code.

We now briefly discuss the challenges related to computational analysis of spatial expression patterns in the manuscript. We added the following sentence in the Discussion: “However, it’s important to note that detection of spatial expression patterns is a challenging computational task, and the performance of analysis methods will depend on the data type (e.g. 1D versus 2D methods) and on the type of spatial patterns (e.g. clearly delimited spatial domains versus smooth gradients).”

7) The scales of the two axes look quite different in Figure 1e. Why?

The difference in scales was caused by different sequencing depths of these two libraries: Replicate 1 was sequenced more deeply than replicate 2, and we here compared reads summed over all sections without any normalization between libraries. We have now added a comment about this in the figure legend.

Reviewer #2 (Remarks to the Author):

Manuscript by Holler et al is a describes a new resource. Authors improved a previously established tomo-seq method to measure RNA localization in one-cell stage zebrafish embryos. They also developed an approach based on 4su metabolic labeling of newly synthesized transcripts in zebrafish embryos and performed single-cell RNA-seq which allowed to predict enrichment of vegetally or animal pole derived transcripts to different cell types. And finally, the authors applied tomo-seq approach to study localization of maternal transcripts in *Xenopus tropicalis* and *Xenopus laevis* species. Motif analysis of 3'UTR sequences identified candidate sites overrepresented in vegetally localized genes.

Overall, the study is well performed, the statistical analysis and conclusions are appropriate, the results are novel and will provide a useful resource for mRNA localization and functional studies. Tomo-seq approach by itself is not novel, and has been used previously. However, using higher resolution and improved analysis approaches, Holler et al identified ten-fold higher number of vegetally localized mRNAs than previously known. Application of 4sUTP labeling for mRNA labeling in vertebrate embryos is novel and results in unique information. However, the major conclusion of this experiment is that vegetally localized genes are enriched in PGCs, which by itself is not particularly novel and has been previously demonstrated using conventional approaches. Evolutionary studies of vegetal mRNAs and 3'UTR element characterization is intriguing. However, it is not clear if any of identified candidate sequences have a functional significance, and this study would be greatly enhanced by functional characterization of some of these 3'UTR elements.

We would like to thank the reviewer for the positive assessment of our work and for the constructive feedback.

Specific points:

1. Supplementary tables would benefit from a brief legend included at the title line which explains annotations in the table.

We thank the reviewer for this comment. We have now added a brief legend for all supplementary tables. Additionally, we added a section in the manuscript in which we summarize the content of the tables.

2. The list of 97 vegetally localized genes should be provided in a separate table, which also includes annotations of which genes have been previously shown to be localized vegetally. A separate table of genes localized to the animal pole should be also provided. It is difficult to extract this data from Supplemental Table 1 in its current form.

We now added a separate list of the 97 vegetally localized genes in zebrafish (supplementary table 2). In this table, we highlighted all genes that, to our knowledge, had already previously been shown to be vegetally localized in zebrafish. There are six genes for which vegetal localization in the zebrafish has been firmly established. However, it's important to note that for some of the remaining 91 localized zebrafish genes, vegetal localization had already previously been reported in *xenopus* (e.g. *trim36*, *rnf38*).

Enrichment of transcripts at the animal pole in zebrafish is less well defined than the vegetal pole: The animal pole is where the embryo forms, and the large majority of transcripts is transported to the animal pole upon fertilization via cytoplasmic streaming. It is therefore likely

that differences in the set of animally localized genes between the replicate experiments reflect small differences in staging rather than real biological differences. We therefore think it would be misleading to explicitly highlight animally localized genes in zebrafish with a separate table. However, in principle this information is available in supplementary table 1 (SOM profiles 1-8).

Furthermore, we now also included a file with lists of all localizing genes (vegetal and animal) in *X. laevis* and *X. tropicalis* oocytes (supplementary table 7).

3. The manuscript argues that 47 vegetally localized genes were significantly enriched in PGCs, and 28 of them were marker genes for PGCs. It is difficult to see this result in Suppl. Table 3 which has marker genes for all clusters listed. The authors should provide a separate table that lists all 47 genes and note fold change in different cell types, and also note which 28 genes were among marker genes for PGCs.

In the new supplementary table 2, we now marked the 47 genes that are expressed in our scSLAM-seq dataset, added their fold change in the respective cell types, and identified marker genes for PGCs.

Also, please check Suppl. Table 3; there are duplicate cluster assignments listed for a subset of genes (*cdx4*, *ved*, *hes6* and others)

Thank you for this comment. As the reviewer correctly noticed, some genes appear multiple times in the table. This is correct – these genes are marker genes for multiple cell types. Due to the relatively low remaining expression of some of these maternal genes, and due to the generally lower discriminative power of the maternal transcripts compared to zygotic transcripts at 6hpf, we used a relatively low cutoff for identification of marker genes, which led to some genes being picked up as marker genes in more than one cluster. This mostly happens for related cell types (i.e. different types of mesodermal cell types). The cutoffs for marker gene detection are now explicitly stated in the legend of the tables.

4. The previous studies noted either very few (Owens et al.) or a large majority of genes (Sindelka et al.) enriched at the animal pole in xenopus embryos. How did the list of animally localized genes identified in the current study compare with the genes identified in the previous studies? It would be helpful to briefly discuss potential reasons for the differences between studies.

We would like to thank the reviewer for this suggestion. There are several potential reasons for discrepancies between our approach and previous publications, such as differences in spatial resolution, normalization, and cutoffs. We now discuss this in the manuscript. Below we compare our results to each of the three previous publications (Owens et al., Sindelka et al., Claussen et al.).

Owens et al.:

The authors sequenced RNA from animal and vegetal caps. After normalization they performed a differential expression analysis, requiring a 4-fold increase for vegetal transcripts and a 10-fold increase for animal transcripts. It is likely that the combination of low spatial resolution and a very strict cutoff led to identification of only 15 animally localized genes. Only one of these 15 genes (*prkag1*) was categorized as “not localized” in our data. However, due to differences in transcriptome annotations, the comparison to our results was not possible for all genes.

Sindelka et al.:

The authors sequenced RNA from parallel aligned oocytes that were dissected into 5 sections. Importantly, the authors did not apply a normalization to the same number of total reads per section. They find that 94% of the genes belong to the category “animal localization”, which is clearly a consequence of their lack of total-read normalization. This does not mean that the analysis of Sindelka et al. is wrong – they are merely asking a different question (absolute spatial patterns) than the other three publications (including ours), which all focus on spatial patterns relative to the majority of genes.

Claussen et al.:

The authors sequenced RNA from animal and vegetal halves and performed differential expression analysis. Importantly, Claussen et al. analyzed both *X. laevis* and *X. tropicalis*. This data should be directly comparable to ours, since none of the discrepancies that we found for Owens et al. (thresholds) and Sindelka et al. (normalization) apply here. We therefore proceeded to compare the data by Claussen et al. in more detail to our results, for *X. laevis* as well as *X. tropicalis*: As summarized in the table below, as well as in Fig. R2-1, we found a considerable overlap in the detected localization patterns. However, there are also some genes that are classified as “animal” or “vegetal” in Claussen et al., but which lie outside the SOM profiles we defined for animal or vegetal localization.

	X. laevis		X. tropicalis	
	Claussen et al.	Tomo-seq	Claussen et al.	Tomo-seq
Vegetally localized	253	214	231	372
Overlap vegetal genes	108		174	
Animally localized	276	270	209	295
Overlap animal genes	90		133	

Fig. R2-1. Comparison of localized RNAs identified by Claussen et al. to SOM analysis of our tomo-seq experiments. Dashed lines: localization cut-offs (faint dashed lines: localization cut-offs from the combination of tomo-seq datasets.)

5. Authors found low evolutionary conservation between vegetally localized genes. A trivial possibility could be that many of these genes showed low expression in some of the species

and were filtered out during the analysis. What was the localization of the 97 zebrafish vegetal genes in other species? Did these genes show different localization pattern, or were they just not detected during analysis in other species?

The reviewer raises an important point. In principle, there could be three possible reasons why a vegetally localized zebrafish gene is not detected as localized in xenopus: i) The gene is indeed not localized. ii) The gene is expressed lowly and is therefore filtered out. iii) The localized zebrafish gene cannot be assigned to a xenopus orthologue. We now checked which of the vegetally localized zebrafish genes are expressed in xenopus. We found 53 of the 97 vegetal zebrafish genes to be expressed above the cutoff in both frog species, and we plotted their localization profiles for both xenopus species in Fig. R2-2. This plot shows that a large fraction of the vegetally localized zebrafish genes are indeed assigned to the “non-localized” category in xenopus. We now explicitly mention the number of 53 out of 97 expressed genes.

Another (more subtle) possible reason for the apparently low conservation of vegetal localization could be that some genes pass the expression cutoffs, but are expressed much lower in xenopus compared to zebrafish. For lowly expressed genes, vegetal localization might possibly be masked by sampling noise. It is difficult to investigate this possibility systematically, since the expression level required for calling a vegetal gene will depend on the exact spatial pattern of the gene. However, we found that the expression levels of the 53 genes are actually higher in xenopus than in zebrafish (Fig. R2-2), which suggests that masking of spatial patterns by sampling noise is probably not a major factor. Nonetheless, we now discuss this possibility in the Discussion.

Fig. R2-2. Left: Spatial expression patterns (SOM categories) in *X. laevis* and *X. tropicalis* for the 53 vegetally localized zebrafish genes that are expressed in both frog species. The region in the bottom left contains the conserved vegetally localized genes. Right: Violin plots for the expression distribution of the 53 vegetally localized zebrafish genes in all three species.

6. The authors describe 9 conserved vegetally localized genes. What was the localization of each of these 9 genes at gastrula stage based on scSLAM-seq analysis?

We would like to thank the reviewer for this suggestion. Only three out of these nine genes were detected above our expression cutoff in the scSLAM-seq data. These genes are: *ppp1r3b*, *dazl* and *camk2g1* (Figure R2-3), and they are enriched in PGCs. We now explicitly mention these genes in the text.

Figure R2-3. Fold change enrichment of maternal transcripts for *ppp1r3b*, *dazl* and *camk2g1* in the different cell types vs. all other cells.

7. What was the localization of *anln* in *X. laevis*? The text says that it was not vegetally localized but it is not clear what its localization pattern was.

The gene *anln.L* in *X. laevis* was ranked in profiles 10 (replicate 1) and 25 (replicate 2) in our SOM analysis. These profiles correspond to genes without a clearly detectable spatial pattern. Figure R2-4 shows the localization pattern for *anln* in *X. laevis*, and the patterns for *dazl* as a comparison. In the text we now specify that we did not detect a localization pattern for *anln* in *X. laevis*.

Fig. R2-4. Localization patterns of *anln* and *dazl* in *X. laevis* (two replicates).

8. Analysis of 3'UTR identified candidate motives enriched in vegetally localized genes. The manuscript would greatly benefit from experimental validation of these motives. Are any of them required or sufficient for vegetal localization? Injection of a reporter mRNA which has these sequences intact or mutagenized could be used to answer this question.

The reviewer raises an important point: We identified candidate motifs and investigated their evolutionary conservation, but we agree with the reviewer that functional experiments would be needed to validate the role of these sequence elements in mRNA localization. The experiment suggested by the reviewer (injection of reporter mRNAs) is intriguing but experimentally challenging: Injection of reporter mRNAs into fertilized eggs is a standard technique and has been used to e.g. study mRNA degradation (Yartseva et al., Nature Methods, 2017; Rabani et al., Mol Cell, 2017). However, for analysis of mRNA localization, injection into immature oocytes would be required, since the localization process is already completed in mature oocytes, and the localization machinery is not active any more in fertilized eggs (Pelegri, Dev Dyn, 2003, Kosaka et al., Mech Dev, 2007). Injection and culturing of immature zebrafish oocytes is not a standard technique. However, inspired by the reviewer's suggestion, we now attempted the following experiment:

As a first pilot experiment, we cultured extracted immature oocytes for 22h without injection (Nair, Pelegri 2013). Live/dead staining (FDA and PI) revealed a mixture of healthy and dying cells (Figure R2-5).

Figure R2-5: Live and dead stain of extracted zebrafish oocytes.

Figure R2-6: Injection of fluorescently labeled mRNA into zebrafish oocytes.

Despite attempts at optimizing the extraction procedure and the culturing conditions, we were unable to obtain a larger fraction of live cells. In a next set of experiments, we injected a reporter construct consisting of the CDS of dTomato and the 3'UTR of the well-characterized vegetally localized gene *dazl* as a positive control (Fig. R2-6 a). The mRNA was fluorescently labeled with Aminoallyl-UTP-ATTO488 to allow direct detection of localization patterns by stereofluorescence microscopy. We injected the construct into oocytes of various sizes in order

to cover oocytes at different stages of maturation. Immediately after injection, the mRNA could be detected as a cloud emanating from the point of injection (Fig. R2-6 b). However, after culturing for 4h and 22h, the fluorescent signal had disappeared from most cells, suggesting that the mRNA had been degraded (Fig R2-6 c,d).

Of note, we detected strong red fluorescence in some cells (Fig. R2-6 d), indicating that the injected mRNA was translated and persisted long enough to produce considerable levels of dTomato protein. In those cells with detectable remaining reporter mRNA, we did not observe a clear localization of the green signal at either 4h or 22h. At this point we decided to stop these experiments, since the most likely reason for the failure of the experiment is suboptimal oocyte culturing conditions, optimization of which is outside our field of expertise and might easily develop into a time-consuming project of its own. However, we fully agree with the reviewer that this is a very interesting research question, which might be suitable for a follow-up project.

Reviewer #3 (Remarks to the Author):

Summary

Holler et al investigated the spatial localization of mRNAs in the single-cell zebrafish embryo and distinguished maternal and zygotic transcripts at gastrulation stage using recently developed sequencing technologies. Using these data, they find that vegetally localized genes in the one cell embryo are enriched in primordial germ cells at gastrulation. Based on previous literature on conservation of 3'UTR elements that may drive transcript localization across species, they asked whether localized genes found in zebrafish are shared in *Xenopus laevis* and *tropicalis* embryos. They identified 9 genes that localize vegetally in zebrafish and two species of *Xenopus* and further investigated sequence elements in 3'UTR of these genes to find repeated motifs that are linked to characterized functions such as RNA stability and motifs not yet described.

This manuscript provides datasets that should be useful for others in the field interested in maternal mRNA localization and turnover, it showcases two relatively new sequencing technologies, and provides some new biological insight into germ cell localized RNA, so I support publication of this manuscript. My comments are aimed mostly at helping the authors properly contextualize and describe their findings.

We thank the reviewer for the positive assessment of our work, and we are grateful for the feedback regarding the presentation and contextualization of our results.

Text

1) After reading the abstract (and indeed the introduction), it is not exactly clear what the authors did experimentally. As this is largely a technique driven paper, I think it is useful to spend a sentence naming and/or describing the tomo-seq and scSLAMseq technique in the abstract and intro. This is especially important as there are new versions and competing approaches constantly being developed, so it would be nice for readers to know early which techniques these were.

We now explicitly mention tomo-seq and scSLAM-seq in the abstract, and we describe these experimental approaches briefly in the introduction.

2) Since there were two timepoints captured - one-cell stage and gastrulation, "dynamics" (including the title) might not be accurate.

This is a good point. We have changed the title to "Spatio-temporal mRNA tracking in the early zebrafish embryo", which we think reflects the experimental approach better. We also changed the wording throughout the manuscript.

3) Introduction could use some reframing to tie in the biological question investigated (RNA transport in early development?) and then lead into the limitations of tools available and how the authors utilized combinations of approaches and developed new ways to interrogate this question.

We have now rewritten the introduction in order to state the biological question (RNA transport in early development) earlier and more clearly, and we now discuss the limitations of current approaches with respect to our experimental system.

4) In results, it would be useful to better justify the orientation and timing chosen for tomo-seq with reference to what is known about vegetally localized mRNAs and microtubule transport. The equivalent of cortical rotation in zebrafish has already happened at 30 minutes, so this is a bit of a late stage to choose for tomo-seq with an animal-vegetal orientation. There is an argument that an earlier stage (5 minutes after fertilization, i.e. before cortical rotation) should have been chosen to identify vegetally localized mRNAs in the egg using tomo-seq along the animal-vegetal axis and then dorsal-ventral tomo-seq at 30 minutes post fertilization.

We agree with the reviewer that it would have been a “cleaner” experiment to perform tomo-seq at 5 minutes post fertilization. Our decision to use the 30 minutes time point was mostly motivated by practical considerations. In particular, we were concerned that we would increase technical error, since it would be difficult to prepare samples at the 5 min stage with the necessary precision due to the time needed for transport and embedding. We now explicitly state this in the Methods.

Furthermore, we expected that the genes involved in ‘cortical rotation’ would still show up as vegetally localized in our data. *Wnt8a* and *grip2a* are two zebrafish genes that are known to undergo the equivalent of cortical rotation. The dorsal shift of these genes is relatively mild at the relevant stages (see Fig. 3c from Ge et al. below). Indeed, these two genes are categorized as vegetally localized in all replicates.

We agree with the reviewer that tomo-seq along the D-V axis would be interesting. We here decided against this option due to the requirement for additional markers to stain the D-V axis. That said, we believe that such an experiment should be feasible, and this would indeed be an interesting follow-up experiment.

4) The authors repeatedly say “sub single cell tomo-seq”. While technically true, the fertilized egg is a very unique cell and tomo-seq in general is not a good method for sub single cell transcriptomics compared to FISH based approaches.

We agree with the reviewer that sub-single-cell resolution by tomo-seq is only possible in special cell types like the fertilized egg. We have rephrased the “sub-single-cell” statements, and we now explain better that sub-single-cell resolution relies on the unique dimensions of the fertilized egg.

5) [lines 177-179] To suggest that *wnt8a* and syntabulin are degraded more rapidly than germ cell factors, further support from literature and/or validation needed, or an argument that their method has the dynamic range to detect such a lowly expressed gene.

The reviewer raises an important concern: Failure to detect lowly expressed genes like *wnt8a* or *syntabulin* might be due to insufficient sequencing depth rather than downregulation of the genes. We therefore now revisited the expression relative to the sequencing depth in more detail: In the tomo-seq data (replicate 1) we detect *wnt8a* and *sybu* with 279 and 219 reads, respectively, out of a total of approx. 19 million mapped reads in valid sections. In the scSLAM-seq data (replicate 1) we do not find any reads for *wnt8a* and *sybu* in the maternal fraction of approx. 16 million unlabeled mapped UMIs in valid single cells. For comparison, we detect 1146 reads for *dazl* in tomo-seq, and 221 UMIs in scSLAM-seq. These numbers illustrate that the dynamic range of our experimental approaches should be sufficient to quantify the stability of these lowly expressed genes. However, we agree with the reviewer that further experiments would be needed to draw any general conclusions from this observation. We apologize that we had not marked the sentence in lines 177-179 more clearly as speculation, and we now removed this statement from the manuscript.

6) The authors should add p-values to the k-mer enrichment analysis of 3'UTRs for localized genes. The identified motifs are short and unclear if they are enriched by chance. Especially, since they are not experimentally tested.

We added e-values of the respective motifs to Fig. 6d and e.

Citations

7) Should cite classic papers on vg1 mRNA localization in *Xenopus* from Melton lab.

We now cite the identification of Vg1 as well as the dorsalizing factor Xwnt-11 as vegetally localized: "...*xenopus* as well as zebrafish use the vegetal pole to store factors for germ cell specification and dorsoventral axis determination (Ku and Melton 1993, Melton 1987)."

Additionally, we now cite the Melton lab for proposing a two-step model for transport and anchoring of mRNA to the vegetal pole (see 8).

8) [line 91] That the transcripts that are enriched in the vegetal pole are "specifically transported and retained" needs citation.

We now added a citation to support our statement: "[...] were specifically transported and retained at the vegetal pole (Yisraeli et al. 1990)."

9) [line 104 and 105] Citation needed to support "..., which increases the number of known vegetal genes by about tenfold" and "...have previously been shown to localize vegetally."

We now added a supplementary table with all zebrafish genes that we identified as vegetally localized, including references for those genes that were already known to be localized (supplementary table 2).

10) [Figure 5a ii] Citation needed for the phylogenetic tree and label figure with species names.

We now added the citation in the figure legend and labeled the picture with species names.

Format/ minor text changes

11) [Figure 2d] Axis labels missing and scale bars missing on images of embryos. Panel could be larger.

We added axis labels and scale bars to Figure 2d. We now also increased the size of the panel as much as the figure layout permits.

12) [Figure 5a iii] Label what the black dots represent or note in figure legend. Not sure whether this panel is necessary?

We now describe that these dots represent vegetally localized mRNA molecules. In Figure 5a iii we would like to emphasize the differences in cell division modes between zebrafish and xenopus. We believe this information is important, so we would prefer to keep this panel.

13) For the supplementary tables, could add captions for what each column represents.

We now added a brief legend to all supplementary tables. Additionally, we added a section in the manuscript in which we summarize the content of the tables.

Reviewers' Comments:

Reviewer #1:

Remarks to the Author:

The revised manuscript has been improved significantly. The authors added several new experimental and computational results with a focus on improving spatial analysis beyond 1D – a major concern in my original review. The added results have provided more and better insights into spatiotemporal dynamics of the early zebrafish embryo. However, two points in my original reviews could benefit from more in-depth studies. Here are my comments.

1) If it is difficult to perform 3D reconstruction (point 3 in the original review), is it possible to integrate spatial images on individual genes and the scRNA-seq data by using computational tools (e.g. SpaOTsc, Nat Commun 11, 2084 (2020). <https://doi.org/10.1038/s41467-020-15968-5>)? Some effort or a discussion along this line will be useful.

2) The authors performed one simple interaction analysis (point 4 in the original review). In the revision, the authors already obtained ligand-receptor pairs of their interests. With just a bit of more effort, for example, using a recently published, user-friendly cell-cell communication R-package, CellChat (Nat Commun 12, 1088 (2021). <https://doi.org/10.1038/s41467-021-21246-9>), the authors can obtain much more insights on cell-cell communications along with more intuitive visualization plots.

Reviewer #3:

Remarks to the Author:

The authors have now made the changes I recommended in the revised manuscript which were mostly to clarify the text rather than new experiments. The manuscript is now easier to understand. I am satisfied with the current manuscript.

I was also asked to comment on the concerns of Reviewer #2.

1. Functional characterization of some 3'UTR elements- in the original manuscript the authors perform motif analysis to identify sequences that are enriched in vegetally localized genes. Functional testing of these sequences would be an obvious improvement to the paper but I did not ask for this in my review because it is technically difficult and I thought it would likely fail (both technically hard to do the experiment and the motifs are small). Classically, motifs have been functionally tested by making reporter RNAs with the motif of interest and injecting them into *Xenopus* oocytes. This approach has been successful for Vg1 localization to the vegetal pole and a similar approach can localize RNAs to the PGCs using different motifs. The authors tried this approach in zebrafish but did not get any convincing data. Culturing *Xenopus* oocytes is difficult and culturing zebrafish oocytes is even harder. Furthermore it is hard to get zebrafish oocytes to go through the correct maturation process in vitro and injecting them can pseudo-activate the eggs. It is thus not surprising that the authors failed with this experimental approach. I think a better approach would have been to take advantage of genetics in zebrafish rather than embryology as done previously in *Xenopus*. The authors could have used crispr to delete the motifs from several genes or they could have made transgenics that express a reporter RNA (driven by a ubiquitous or oocyte specific promoter) with the added motif and then performed in situ to test for localization. Endogenously expressed RNA would be better to assess localization as the fluorescently labeled RNA tried by the authors would likely not be processed the same.
2. Analysis of localization of 97 zebrafish vegetal genes in the two *Xenopus* species and analysis of the 9 conserved vegetally localized genes- the authors have now performed the requested analysis and modified the text accordingly. The results do not change the message of the paper but they are an obvious question that readers might have so it is good to include these results.

Reviewer #1 (Remarks to the Author):

The revised manuscript has been improved significantly. The authors added several new experimental and computational results with a focus on improving spatial analysis beyond 1D – a major concern in my original review. The added results have provided more and better insights into spatiotemporal dynamics of the early zebrafish embryo. However, two points in my original reviews could benefit from more in-depth studies. Here are my comments.

We would like to thank the reviewer for the positive feedback on the changes we introduced in the revised manuscript. Below we address the remaining concerns of the reviewer in greater depth. All changes made during the second revision are highlighted in **yellow** in the manuscript.

1) If it is difficult to perform 3D reconstruction (point 3 in the original review), is it possible to integrate spatial images on individual genes and the scRNA-seq data by using computational tools (e.g. SpaOTsc, Nat Commun 11, 2084 (2020). <https://doi.org/10.1038/s41467-020-15968-5>)? Some effort or a discussion along this line will be useful.

The reviewer raises an interesting question: Can we apply recently developed computational methods for integration of spatial information with scRNA-seq to our dataset? Approaches for reconstruction of spatial transcriptomics data based on scRNA-seq and "landmark" genes have made enormous progress since the early publications by Satija et al. and Achim et al. (Nat Biotech, 2015). Specifically, novoSpaRc (Nitzan et al., Nature, 2019) and SpaOTsc (Cang et al., Nat Comms, 2021) have used optimal transport theory for improved mapping. The question is, can these approaches be used to refine spatial patterns of localized genes by combining our tomo-seq data with *in situ* images for some selected genes? We believe that this may indeed be possible, but there are two important caveats to consider:

1) This approach would only be useful if it allows us to gain additional insights by e.g. discovering novel spatial patterns, or by assigning genes to one out of several possible spatial patterns. However, we found rather low complexity of spatial patterns in the one-cell stage embryo by *in-situ* hybridization: All the vegetally localized genes that we analyzed by whole-mount *in-situ* hybridization in Fig. 2d show a simple gradient from the vegetal pole to the animal pole, with the main distinguishing factors being the overall expression level and the background of unlocalized transcripts. In the absence of more complex spatial patterns, it is questionable how much can be gained by computational approaches for spatial data integration.

2) One major difference between our data and typical applications for spatial integration analysis is that we work in a sub-single-cell system. This means that we cannot use the coupling between individual genes that single-cell transcriptomics provides: In scRNA-seq, the spatial expression pattern of one landmark gene can be used to predict the spatial patterns of other co-expressed genes. This is not possible in our data, where all transcripts are located within the same single cell, so it's not clear *a priori* how information about the localization pattern of one gene can be used to predict localization of other genes. That said, it's likely that different types of couplings between individual genes exist in our data. This might for instance be due to shared localization elements that would be predictive of localization patterns. However, this would require additional mechanistic knowledge that is currently not available.

There is hence no straightforward way to integrate spatial images of selected genes to improve predicted patterns for other genes. However, this is an intriguing concept, which we now included in the Discussion, together with a reference to the article by Qing Nie suggested by the reviewer. We added the following paragraph:

“Several methods have been published that pair scRNA-seq data with spatial imaging data, including two novel approaches using optimal transport theory (Nitzan et al., Cang et al.). This raises the intriguing possibility that it might be possible to integrate a limited number of *in-situ* images with our tomo-seq data to predict spatial patterns transcriptome-wide with higher resolution. However, one important difference of our sub-single-cell system compared to scRNA-seq is that we cannot directly use co-expression in single-cells as a coupling between genes. Hence, other types of couplings, e.g. shared localization elements between genes, would be required.”

2) The authors performed one simple interaction analysis (point 4 in the original review). In the revision, the authors already obtained ligand-receptor pairs of their interests. With just a bit of more effort, for example, using a recently published, user-friendly cell-cell communication R-package, CellChat (Nat Commun 12, 1088 (2021). <https://doi.org/10.1038/s41467-021-21246-9>), the authors can obtain much more insights on cell-cell communications along with more intuitive visualization plots.

We agree with the reviewer that analysis of ligand-receptor interactions is an interesting question. Analysis of cell-cell interactions based on gene expression analysis is a rapidly advancing field, with recent improvements regarding statistical analysis of interactions as well as expansion beyond classical ligand-receptor interactions. We had used CellPhoneDB, which is probably the most widely used tool for ligand-receptor analysis. Following the reviewer’s suggestion, we now repeated the analysis with the CellChat method by Qing Nie’s lab. While the two approaches are conceptually very similar, there are also some important differences, such as: 1) The ligand-receptor databases are different, which may lead to some differences in detected interactions. 2) While CellPhoneDB is limited to analyzing specific ligand-receptor interactions, CellChat can aggregate this information into pathway activation profiles. While the latter is advantageous for data visualization, this is only of limited use for our project, since our approach is gene-centric (Maternal or zygotic? Localized or not?) instead of pathway-centric. Fig. R1 summarizes the results we obtained with CellChat:

Figure R1: CellChat analysis. Left: Number of detected interactions between different cell types in our 6 hpf dataset. Right: Weighted interaction strength.

We used the same approach for conversion of human genes to orthologous zebrafish genes as for CellPhoneDB, and we included all genes independent of whether they are mainly maternally or zygotically expressed. In summary, we detect many interactions between different cell types. However, the PGCs, which are of particular interest because of their high fraction of maternal transcripts, do not exhibit any interactions with other cell types. Fig. R2 includes the genes involved in the individual ligand-receptor interactions, and in Fig. R3 we quantify the maternal contribution to these genes.

Figure R2: CellChat analysis. Genes involved in detected ligand-receptor interactions.

Figure R3: CellChat analysis. Fraction of maternal transcripts.

In summary, we detect less ligand-receptor interactions than by CellPhoneDB (see Fig. S5b, c). Specifically, the interaction between LAMP1 – FAM3C, which we found to be enriched in PGCs by CellPhoneDB, is not detected by CellChat. This is because potentially important genes

like LAMP1, AM3C and CXADR are not in the CellChat database. The ligands and receptors that underlie the detected interactions are mostly zygotic, and hence not directly related to the core message of our manuscript. Because of these observations, we decided to stick to CellPhoneDB. While we agree with the reviewer that the visualization tools in CellChat are superior to those of CellPhoneDB, we wish to point out that highly aggregated plots like those in Fig. R1 and R2 are of limited use in our case, since it is difficult to include information about maternal vs zygotic expression. Hence, we believe that the data representations we had chosen in Fig. S5b, c are more appropriate to our specific scientific question about the role of vegetally localized maternal transcripts.

The reviewer asked us to obtain “more insights on cell-cell communications”. Since the reviewer did not specify further which insights these might be, we can only speculate which types of analysis the reviewer would want to see. In principle it would of course be possible to follow up on individual ligand-receptor interactions that we identified in our 6 hpf dataset, or to interpret detected cell-cell communication in the context of known developmental principles. However, we feel a general cell-cell communication analysis in the gastrulation stage zebrafish embryo would be out of the scope of our manuscript. Our manuscript is focused on the role of maternal transcripts that exhibit an intracellular spatial localization pattern in the one-cell stage embryo. Of the 97 vegetally localized maternal genes, 47 genes are still detectable at considerable levels at 6 hpf. None of these genes are involved in ligand-receptor interactions as detected by CellChat or CellPhoneDB. This suggests that the role of maternal vegetal genes is mostly related to cell specification (in particular specification of PGCs), and not so much to cell-cell communication. While it is indeed interesting to expand ligand-receptor analysis to other (non-localized) maternal genes, as we have done in Fig. S5b, c, we think it would distract from the core message of our manuscript to broaden the ligand-receptor analysis further. In particular, other existing datasets (e.g. Farrell et al., Science, 2018 or Wagner et al., Science, 2018) with more cells and more stages would be better suited for a general ligand-receptor analysis. We agree with the reviewer that the argumentation was unclear in the previous version of the manuscript, and we updated the corresponding paragraph in the manuscript as follows:

“Furthermore, we investigated the potential involvement of maternal factors in cell-cell interactions. To this end, we identified ligand-receptor pairs between cell types in our scSLAM-seq data (Methods). We found that none of the remaining 47 vegetal genes are involved in annotated ligand-receptor interactions, suggesting that the role of maternal vegetal genes is mostly related to cell specification (in particular specification of PGCs), and not so much to cell-cell communication. When we expanded the analysis to all maternal genes, we found some maternal ligands and receptors in PGCs, while the potential interaction partners are in a variety of cell types (Fig. S5).”

Reviewer #3 (Remarks to the Author):

The authors have now made the changes I recommended in the revised manuscript which were mostly to clarify the text rather than new experiments. The manuscript is now easier to understand. I am satisfied with the current manuscript.

We would like to thank the reviewer for the positive feedback on the changes we introduced in the revised manuscript.

I was also asked to comment on the concerns of Reviewer #2.

1. Functional characterization of some 3'UTR elements- in the original manuscript the authors perform motif analysis to identify sequences that are enriched in vegetally localized genes. Functional testing of these sequences would be an obvious improvement to the paper but I did not ask for this in my review because it is technically difficult and I thought it would likely fail (both technically hard to do the experiment and the motifs are small). Classically, motifs have been functionally tested by making reporter RNAs with the motif of interest and injecting them into *Xenopus* oocytes. This approach has been successful for Vg1 localization to the vegetal pole and a similar approach can localize RNAs to the PGCs using different motifs. The authors tried this approach in zebrafish but did not get any convincing data. Culturing *Xenopus* oocytes is difficult and culturing zebrafish oocytes is even harder. Furthermore it is hard to get zebrafish oocytes to go through the correct maturation process in vitro and injecting them can pseudo-activate the eggs. It is thus not surprising that the authors failed with this experimental approach. I think a better approach would have been to take advantage of genetics in zebrafish rather than embryology as done previously in *Xenopus*. The authors could have used crispr to delete the motifs from several genes or they could have made transgenics that express a reporter RNA (driven by a ubiquitous or oocyte specific promoter) with the added motif and then performed in situ to test for localization. Endogenously expressed RNA would be better to assess localization as the fluorescently labeled RNA tried by the authors would likely not be processed the same.

We thank the reviewer for these comments. While we believe that genetics approaches are outside the scope for this revision, we are highly interested in following the strategy outlined by the reviewer in the future.

2. Analysis of localization of 97 zebrafish vegetal genes in the two *Xenopus* species and analysis of the 9 conserved vegetally localized genes- the authors have now performed the requested analysis and modified the text accordingly. The results do not change the message of the paper but they are an obvious question that readers might have so it is good to include these results.

Indeed, we agree with reviewers 2 and 3 that inclusion of this information may be interesting for the readers of the article.

Reviewers' Comments:

Reviewer #1:

Remarks to the Author:

I appreciate the author's effort in addressing the two points in my report.

For the point 2), the author made strong effort with some additional analysis that were, however, only included in the response letter. Adding a couple of sentences in Discussion on the observations based on CellChat analysis in Fig. R1-R3 and its contrast with CellPhoneDB could be beneficial for the readers.

Reviewer #1 (Remarks to the Author):

I appreciate the author's effort in addressing the two points in my report.

For the point 2), the author made strong effort with some additional analysis that were, however, only included in the response letter. Adding a couple of sentences in Discussion on the observations based on CellChat analysis in Fig. R1-R3 and its contrast with CellPhoneDB could be beneficial for the readers.

We thank the reviewer for the positive feedback on the changes we introduced in the revised manuscript. We agree that the CellChat analysis may be beneficial to the readers, and we have now included it in the manuscript. The results are shown in the new Fig. S5, and we have changed the description of the ligand-receptor analysis by adding the paragraph below (highlighted in yellow in the manuscript):

“Furthermore, we investigated the potential involvement of maternal factors in cell-cell interactions. We first identified ligand-receptor pairs between cell types in our scSLAM-seq data using two different computational methods (CellChat and CellPhoneDB, see Methods). In both approaches we found that none of the remaining 47 vegetal genes are involved in annotated ligand-receptor interactions, suggesting that the role of maternal vegetal genes is mostly related to cell specification (in particular specification of PGCs), and not so much to cell-cell communication. When analyzing all genes, CellChat identified several potential cell-cell interactions. However, expression of the involved ligands and receptors was mostly zygotic, and we did not detect any interactions between PGCs and other cell types (Supplementary Fig. 5). While the results of our CellPhoneDB analysis were generally similar, we additionally also observed several maternal ligands and receptors in PGCs, with potential interaction partners in a variety of cell types (Supplementary Fig. 6). This discrepancy between the two computational approaches is probably due to differences in the underlying ligand-receptor databases.”